# Understanding Knowledge Integration in Language Models with Graph Convolutions

## Abstract

Pretrained language models (LMs) are not very good at robustly capturing factual knowledge. This has led to the development of a number of knowledge integration (KI) methods which aim to incorporate external knowledge into pretrained LMs. Even though KI methods show some performance gains over vanilla LMs, the efficacy and limitations of these methods are not well-understood. For instance, it is unclear how and what kind of knowledge is effectively integrated into LMs and if such integration may lead to catastrophic forgetting of already learned knowledge. In this paper, we revisit the KI process in an information-theoretic view and show that KI could be interpreted using a graph convolution operation. We propose a simple probe model called *Graph Convolution Simulator* (GCS) for interpreting knowledge-enhanced LMs and exposing what kind of knowledge is integrated into these models. We conduct experiments to verify that our GCS model can indeed be used to correctly interpret the KI process, and we use it to analyze two typical knowledge-enhanced LMs: K-Adapter and ERNIE. We find that only a small amount of factual knowledge is captured in these models during integration. While K-Adapter is better at integrating simple relational knowledge, complex relational knowledge is integrated better in ERNIE. We further find that while K-Adapter struggles to integrate time-related knowledge, it successfully integrates knowledge of unpopular entities and relations. Our analysis also show some challenges in KI. In particular, we find simply increasing the size of the KI corpus may not lead to better KI and more fundamental advances may be needed.

## 1 Introduction

Pretrained language models (LMs) have achieved state-of-the-art performance across various natural language processing (NLP) tasks. Previous works have shown that linguistic knowledge is captured quite well by LMs and it plays a vital role in their success (Liu et al., 2019a; Jawahar et al., 2019). However, factual knowledge is sparse and is expressed in varied ways in text. Thus, LMs are much worse in capturing factual knowledge about the world (Petroni et al., 2019; Wang et al., 2021b). This has led to the development of a variety of knowledge integration (KI) methods which aim to integrate external knowledge into LMs (Colon-Hernandez et al., 2021; Wang et al., 2021a; Zhang et al., 2019). Even though knowledge-enhanced LMs perform better on knowledge-related tasks, we lack a deep understanding about the inner workings of these models. Better downstream performance indicates that some new knowledge has been integrated, but how much knowledge has been successfully integrated, which type of knowledge is integrated is not well-understood.

To understand what knowledge is learned in LMs, many model-agnostic methods have been proposed. Previous works have focused on designing simple classifiers as probe models (Hewitt & Manning, 2019; Ribeiro et al., 2016). To get more reliable interpretation, information-theoretic approaches have also been introduced (Guan et al., 2019; Pimentel et al., 2020; Hou & Sachan, 2021). However, factual knowledge is typically organized as large-scale sparse knowledge graphs (KGs). Previous interpretation methods while being suitable for small linguistic graphs, cannot provide reasonable interpretations for large sparse KGs. Prompting is yet another way to understand what factual knowledge do these models learn. Prompts can be designed to let LMs solve fill-in-the-blanks problems, and the prompt performance can be interpreted as a probe (Petroni et al., 2019; Shin et al., 2020; Zhong et al., 2021). However, these methods rely on manually constructed templates which is very time-consuming.

There are two other common challenges to understand KI. First, there are a large number of approaches for KI. KI in LMs can be implemented by matching sentences to entities or triples in knowledge graphs – called entity-wise integration (Peters et al., 2019; Zhang et al., 2019) and triple-wise integration (Liu et al., 2020; Wang et al., 2021a). There are several modeling choices (Colon-Hernandez et al., 2021) for KI, including proposing some modifications to the Transformer architecture (Peters et al., 2019; Zhang et al., 2019; Liu et al., 2020), verbalizing knowledge triples and using data augmentation for finetuning (Agarwal et al., 2021), and designing objective functions that predict the factual knowledge (Yao et al., 2019; Wang et al., 2021a). How to design a general method to understand KI in both entity-wise and triple-wise manner is challenging. Furthermore, KI is typically implemented in a continual learning setup (Parisi et al., 2019) – KI is usually a secondary pretraining or finetuning step (Lu et al., 2021). As new knowledge is integrated, old knowledge could be catastrophically forgotten (CF; Kirkpatrick et al., 2016). KI could also lead to a situation called catastrophic remembering (CR; Kaushik et al., 2021), where the old knowledge may prevent the integration of new knowledge. Our understanding of these issues is limited.

In this paper, we first revisit the KI process (§2). We formulate KI in an information-theoretic view (§2.1), and construct a transformation to approximate general KI process (§2.2). Then, we prove that the KI process can be simulated by graph convolution operations (§2.3). Second, we introduce how to analyze KI, CR, and CF. Specifically, we prove that the KI process can be interpreted by graph attention mechanism (§3.1). Based on that, we propose Graph Convolution Simulator (GCS) model to simulate and interpret the KI process (§3.2), and introduce the way to analyze its interpretation results (§3.3). We show that our designed probe model GCS can correctly simulate and interpret the KI process for two popular knowledge-enhanced LMs: K-Adapter (Wang et al., 2021a) and ERNIE (Zhang et al., 2019). We find that K-Adapter integrates simple relational knowledge (i.e., entities are leaf nodes in KGs) well, while ERNIE is better at integrating complex relational knowledge (i.e., entities are center nodes in KGs). In our qualitative study, we find that K-Adapter does not learn temporal knowledge at all. We further break down our analysis of KI in terms of type of relations and the popularity of entities. We find that catastrophic remembering often happens to simple relational knowledge (i.e. simple relational knowledge is harder to edit), while complex relational knowledge is often catastrophically forgotten. We also find that catastrophic forgetting easily happens to popular entities, while catastrophic remembering often happens when the entities are not very common. Finally, we investigate the correlation between the size of the KI corpus and KI quality. We find that there is no apparent positive relationship between them, suggesting that merely building larger KI datasets may not be enough and we may need to make more fundamental advances to build better knowledge-enhanced language models.

## 2 SIMULATING KNOWLEDGE INTEGRATION

In this section, we revisit KI in an information-theoretic view, and prove that KI can be simulated by graph convolutions. Specifically, we first formulate the KI process. We use MI to measure the knowledge learned in LMs, and use the change of MI to define KI, catastrophic remembering (CR), and catastrophic forgetting (CF). Then, based on the definition, we construct a multistep transformation to approximate the KI process with arbitrary accuracy. We show that KI can only happen in certain steps of on the transformation, which are graph convolutions.

### 2.1 KNOWLEDGE INTEGRATION DEFINITION

Before presenting a formal definition of KI based on MI, we introduce some basic concepts.

**Knowledge graphs.** We assume that factual knowledge can be formulated as a knowledge graph $\mathcal{G} = (\mathcal{V}, \mathcal{E})$, where nodes $v_i \in \mathcal{V}$ represent entities, and edges in $\mathcal{E}$ represent relations between them. Let $\mathcal{N}_{v_i}$ denote the set of neighbors of node $v_i$, and $t_i$ denote the entity label corresponding to the node $v_i$. Further, let $\boldsymbol{x}_i = \text{LM}(t_i)$ denote the entity (label) representations of $t_i$ given by a LM[1]. Let $\boldsymbol{X} \in \mathbb{R}^{|\mathcal{V}| \times d}$ denote a matrix formed by stacking all entity representations $\boldsymbol{x}_i \in \mathbb{R}^d$. In this paper, we only consider nodes and relations in the KG and ignore other kinds of KG information such as edge weights, edge directions and multi-edges (multi-relations).

---

[1]We represent each entity as the average of its word(-piece) embeddings given by the LM as Hewitt & Manning (2019) and Hou & Sachan (2021).

**Information-theoretic probe.** We follow theoretical settings of Hou & Sachan (2021) that measures the knowledge captured in LMs by MI. We assume that the local graph structure $\mathcal{G}(v_i)$ contains all the factual information regarding $v_i$. In this work, we consider factual knowledge in the form of triples $(v_i, r, v_j)$. Since a triple only contains entities within one-hop, it suffices to set $\mathcal{G}(v_i) = \mathcal{N}_{v_i}$. Factual knowledge that has been successfully integrated should be reflected in entity representations. Let $\mathbf{x}$ be a random variable that takes values ranging over all possible entity representations of a LM[2], and $\mathbf{g}$ be a random variable that ranges over all possible corresponding local structures $\mathcal{G}(v_i)$. Intuitively, $\mathrm{MI}(\mathbf{x}; \mathbf{g})$ measures the amount of information in $\mathbf{g}$ that is contained in $\mathbf{x}$.

**Definition 1** (Knowledge Integration). *Given $\mathcal{G}$, let entity representation matrices given by a LM before and after KI be $\boldsymbol{X}$ and $\boldsymbol{H}$. Corresponding random variables are $\mathbf{g}$, $\mathbf{x}$, and $\mathbf{h}$. We formulate KI process $f(\mathbf{x}, \mathbf{g}) = \mathbf{h}$ as the change of MI: $\mathrm{MI}(\mathbf{x}; \mathbf{g}) \rightarrow \mathrm{MI}(\mathbf{h}; \mathbf{g})$, i.e., knowledge of $\mathbf{g}$ is integrated into $\mathbf{x}$ to get $\mathbf{h}$. If $\mathrm{MI}(\mathbf{h}; \mathbf{g}) \approx \mathrm{MI}(\mathbf{x}; \mathbf{g})$, it means that CR happens, i.e., there is a failure to integrate new knowledge. If CF happens, we have $\mathrm{MI}(\mathbf{h}; \mathbf{x}) \approx 0$.*

The definition can be intuitively visualized by Figure 1. The MI change can be represented by regions 1 and 4 in the Venn diagram. Ideally, if we have $\mathrm{MI}(\mathbf{h}; \mathbf{g}) - \mathrm{MI}(\mathbf{x}; \mathbf{g}) \approx \mathrm{MI}(\mathbf{g}; \mathbf{g}) - \mathrm{MI}(\mathbf{x}; \mathbf{g})$, i.e., the region 5 is small, we say most knowledge is successfully integrated. If little new knowledge has been integrated, the region 4 is very small. Then, we say that CR has happened. If CF happens, most knowledge in $\mathbf{x}$ is forgotten in $\mathbf{h}$ after KI, and region 1 is large. Successful KI happens when much new knowledge is integrated (i.e., $\mathrm{MI}(\mathbf{h}; \mathbf{g})$ is large) and little old knowledge is forgotten (i.e., $\mathrm{MI}(\mathbf{h}; \mathbf{x})$ is large).

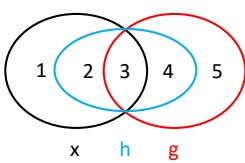

**Figure 1:** Venn diagram for MI visualization. $\mathbf{x}$, $\mathbf{h}$, and $\mathbf{g}$ are random variables.

## 2.2 APPROXIMATED TRANSFORMATION CONSTRUCTION

In this subsection, we show that we can construct a transformation to approximate the KI process with arbitrary accuracy. We begin by introducing the concept of Graph Fourier transforms.

**Graph Fourier transformation.** Graph Fourier transform (GFT) can be used to transform the entity representation matrix $\boldsymbol{X}$ in the Euclidean space to the graph spectral domain (i.e., KG space). Specifically, let $\boldsymbol{A} \in \mathbb{R}^{|\mathcal{V}| \times |\mathcal{V}|}$ be the symmetric adjacency matrix corresponding to $\mathcal{G}$. Let $\boldsymbol{L}_n = \boldsymbol{I} - \boldsymbol{D}^{-1/2} \boldsymbol{A} \boldsymbol{D}^{-1/2}$ denote the normalized Laplacian matrix for $\mathcal{G}$, where $\boldsymbol{D}$ denotes the degree matrix of $\mathcal{G}$. We do the eigendecomposition for $\boldsymbol{L}_n$ as $\boldsymbol{L}_n = \boldsymbol{U} \boldsymbol{\Lambda} \boldsymbol{U}^T$, where $\boldsymbol{U}$ is the matrix of eigenvectors ordered by eigenvalues and $\boldsymbol{\Lambda} = \mathrm{diag}(\lambda_1, \lambda_2, ..., \lambda_N)$ is the diagonal matrix of eigenvalues. Based on the GFT (Sandryhaila & Moura, 2014), formally, the transformation of $\boldsymbol{X}$ to the KG space can be written as $\mathrm{GFT}(\boldsymbol{X}) = \boldsymbol{U}^T \boldsymbol{X}$, and its inverse transformation can be written as $\mathrm{RGFT}(\mathrm{GFT}(\boldsymbol{X})) = \boldsymbol{U}\mathrm{GFT}(\boldsymbol{X}) = \boldsymbol{X}$.

It is intractable to directly simulate the KI process (with simple probe models), i.e., integrating knowledge information from $\mathbf{g}$ into $\mathbf{x}$ to get $\mathbf{h}$ (Definition 1), since $\mathbf{g}$ is graph data that are not in the Euclidean space. Fortunately, with the help of GFT, we can transform $\mathbf{x}$ and $\mathbf{h}$ into the KG space, and design transformation there to simulate the KI process. Below theorem provides a feasible way to construct a transformation that can simulate the KI process with arbitrary accuracy.

**Theorem 2** (Transformation Existence). *Denote the graph Fourier transformation and its inverse transformation in terms of $\mathcal{G}$ as $\mathrm{GFT}(\cdot)$ and $\mathrm{RGFT}(\cdot)$. Given a LM and its knowledge-enhanced version, suppose that $\mathrm{MI}(\mathbf{h}; \mathbf{g}) - \mathrm{MI}(\mathbf{x}; \mathbf{g}) > 0$, and there exists a mapping that satisfies $f(\mathbf{x}, \mathbf{g}) = \mathbf{h}$. Then, for any $\epsilon > 0$, there exists a neural network $\mathrm{NN}(\cdot)$ such that*

$$|f(\mathbf{x}, \mathbf{g}) - \mathrm{RGFT}(\mathrm{NN}(\mathrm{GFT}(\mathbf{x})))| < \epsilon. \tag{1}$$

The proof can be found in Appendix B. Theorem 2 shows that there exists an approximated transformation composed of GFT and a neural network that can simulate the KI process (i.e., $f(\mathbf{x}, \mathbf{g}) = \mathbf{h}$) with arbitrary accuracy. In practice, the mapping $f(\mathbf{x}, \mathbf{g})$ is realized by complex KI training process on large LMs with new datasets, objectives, or even new LM parameters. Here, we can use a simple transformation $\mathrm{RGFT}(\mathrm{NN}(\mathrm{GFT}(\mathbf{x})))$ to generally approximate and simulate the complex KI process. However, KGs are normally very large, and computing the eigendecomposition of

---

[2]Here, the set of entity representations $\boldsymbol{X} \in \mathbb{R}^{|\mathcal{V}| \times d}$ can be regarded as empirical samples from $\mathbf{x}$.

the Laplacian matrix for GFT is prohibitively expensive. Besides, the transformation is not interpretable, which can not be used as a probe model directly. Thus, we take a deeper look into this transformation. We first show that it can be further simplified as graph convolutions.

## 2.3 KNOWLEDGE INTEGRATION SIMULATION WITH GRAPH CONVOLUTIONS

In this subsection, we introduce details of the transformation in Theorem 2. We prove that even if there are multiple steps of the transformation, MI change only happens in certain steps. We show that these steps are equivalent to graph convolutions.

**Graph convolutions.** Convolution operations on graphs are often used to model relational information of KGs, where entities aggregate information from their neighbors and pass the information along based on the graph structure. Graph convolutions can be implemented by filters $g_\Theta$ in the graph spectral domain (i.e., KG space). As the GFT of the convolution of $g_\Theta$ and $\boldsymbol{X}$ is the pointwise product of their GFT (Bracewell & Bracewell, 1986), the convolution can be written as $g_\Theta \star \boldsymbol{X} = \mathrm{RGFT}(g_\Theta \cdot \mathrm{GFT}(\boldsymbol{X}))$ (Bruna et al., 2014).

In the below proposition, we show that the multistep transformation for KI simulation (Theorem 2) can be simplified as graph convolutions.

**Proposition 3** (Graph Convolutions for Simulation). *Suppose* $\mathrm{MI}(\mathbf{x}; \mathbf{g}) < \mathrm{MI}(\mathbf{h}; \mathbf{g})$, $\mathrm{MI}(\mathbf{h}; \mathbf{x}) < \mathrm{MI}(\mathbf{x}; \mathbf{x})$, *and the mapping* $f(\mathbf{x}, \mathbf{g}) = \mathbf{h}$ *can be well approximated by the transformation* $\mathrm{RGFT}(\mathrm{NN}(\mathrm{GFT}(\mathbf{x})))$ *where the neural network* $\mathrm{NN}(\cdot)$ *has* $n$ *layers. We can only use* $n$ *linear functions in the neural network (i.e., graph convolutions) to well simulate the MI change.*

Proposition 3 indicates that if some new knowledge is integrated and some old knowledge is forgotten during KI, we can only use $n$ linear functions instead of the whole transformation in Theorem 2 to simulate the MI change (i.e., formulated KI process). These linear functions in the KG space are actually graph convolution operations. And thus, we can circumvent the expensive eigendecomposition in GFT with fast and well-approximated graph filters (Defferrard et al., 2016; Kipf & Welling, 2017). The formal proof is in Appendix C. Figure 2 briefly illustrates the overall idea.

The black dashed transformation proposed in Theorem 2 can simulate KI with arbitrary accuracy. Even if the transformation is simple, it is still not very efficient with large KGs and not interpretable. The red solid lines represent the simulation of KI using graph convolutions (Proposition 3), which are much more fast and interpretable.

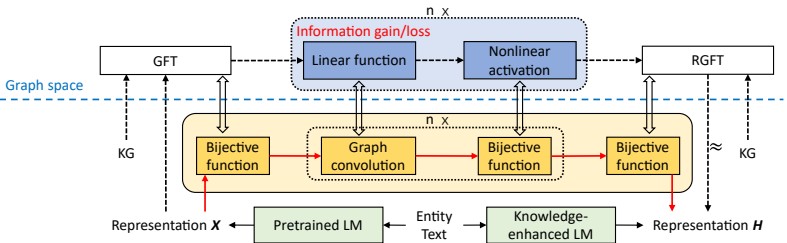

**Figure 2:** Illustration of the KI simulation. Black dashed arrows show the approximated transformation in Theorem 2. Red arrows show the simplified simulation with graph convolutions in Proposition 3.

According to the invariance property of MI (Kraskov et al., 2004), the introduction of bijective functions does not introduce any new information – MI remains unchanged upon the introduction of bijective functions. We know that GFT and RGFT are both bijective (Appendix C.1). We show that nonlinear activation functions in a neural network (e.g., $\mathrm{sigmoid}(\cdot)$) are bijective as well (proof in Appendix C.1). Thus, the MI change in the KI process can only happen in the linear function (proof in Appendix C.2). Based on the convolution theorem (Bracewell & Bracewell, 1986), linear functions in graph space are graph convolutions (Sandryhaila & Moura, 2014; Bruna et al., 2014; Kipf & Welling, 2017) (Appendix C.3). Thus, we can simply use graph convolution operations instead to simulate the KI process.

## 3 INTERPRETING KNOWLEDGE INTEGRATION

In the last section, we showed that KI can be simulated by graph convolutions. In this section, we introduce the mechanism to interpret the KI process. Specifically, we first illustrate that the

graph attention mechanism can be used to interpret graph convolutions. Then, we introduce an implementation of the probe model proposed in our work: Graph Convolution Simulator (GCS), as well as a method to analyze our interpretation results.

## 3.1 Knowledge Integration with Graph Attention

Following Proposition 3, we can select powerful graph filters to simulate and interpret the KI process. Velickovic et al. (2018) and Thekumparampil et al. (2018) introduce the attention mechanism to graph filters, where the contribution of each edge to the convolution can be shown explicitly. Graph attention makes filters more powerful and convolutions more interpretable (Fu et al., 2020).

**Proposition 4** (Graph Attention for Interpretation). *Suppose* $\mathrm{MI}(\mathbf{x}; \mathbf{g}) < \mathrm{MI}(\mathbf{h}; \mathbf{g})$, $\mathrm{MI}(\mathbf{h}; \mathbf{x}) < \mathrm{MI}(\mathbf{x}; \mathbf{x})$, *and the KI process* $\mathrm{MI}(\mathbf{x}; \mathbf{g}) \to \mathrm{MI}(\mathbf{h}; \mathbf{g})$ *can be well simulated by* $n$ *graph convolution operations. Then, we can use the attention coefficients on edges and self-loops to interpret the KI, CR, and CF.*

The formal proof can be found in Appendix D. Note that we have $n$ graph convolutions, and they function differently (Bruna et al., 2014). In a multi-layer graph convolution network, the $k$-th graph convolution step aggregates information from $k$-hop neighbors[3]. In this work, we consider integration of knowledge in the form of knowledge triples. Knowledge triples link entities within 1-hop. Thus, in practice, we set $n = 1$ for simplicity in our work. According to Fu et al. (2020), graph attention can also be seen as edge denoising. This provides an alternative explanation of our GCS probe from the denoising view. More details can be found in Appendix D.

## 3.2 GCS Architecture.

Based on Proposition 3 and Proposition 4, we design GCS with two bijective functions and one graph convolution function in between. To implement a bijective function in practice, we show that special MLP layers can be bijective if the weight matrix is a square matrix (plus a small noise). The formal description and the proof are in Appendix E. We design our GCS model with one graph convolutional layer and two bijective MLP layers as:

$$\mathrm{GCS}_{\theta_1}(\cdot) = \mathrm{MLP}_n(\mathrm{GC}(\mathrm{MLP}_n(\cdot), \mathcal{G})), \tag{2}$$

where $\mathrm{MLP}_n(\cdot)$ is the bijective MLP layer and $\mathrm{GC}(\cdot, \mathcal{G})$ is the graph convolutional layer on the KG $\mathcal{G}$ which is used to simulate and interpret the KI process. Given an entity $v_i$ and its set of neighbors $\mathcal{N}_{v_i}$, we can write the graph convolutional layer as:

$$\mathrm{GC}(\boldsymbol{x}_i) = \sigma \left( \sum_{v_j \in \mathcal{N}_{v_i} \cup \{v_i\}} a_{i,j} \boldsymbol{W}^V \boldsymbol{x}_j \right), \text{ where } a_{i,j} = \mathrm{softmax}\left( \frac{(\boldsymbol{W}^Q \boldsymbol{x}_i) \cdot (\boldsymbol{W}^K \boldsymbol{x}_j)}{\sqrt{d_k} \cdot t} \right). \tag{3}$$

Here, $\boldsymbol{x}_i$ is the entity representation of $v_i$ before knowledge integration, the activation function $\sigma(\cdot)$ is $\mathrm{ELU}(\cdot)$ function, and $\boldsymbol{W}^V$ is a weight matrix. $a_{i,j}$ is the attention coefficient on the edge that connects $v_i$ and $v_j$. $\boldsymbol{W}^Q$ and $\boldsymbol{W}^K$ are two parameter matrices in the graph attention. $d_k$ is the dimension of vector $\boldsymbol{W}^K \boldsymbol{x}_j$, and $\mathrm{softmax}(\cdot)$ is the edge-wise softmax function with respect to node $v_i$. Temperature $t$ is a hyperparameter that controls the attention distribution to be hard or soft. As the multiplicative attention mechanism (Vaswani et al., 2017) is broadly used in LMs, we also select multiplicative attention in the graph attention. We optimize GCS by letting its outputs be as close to $\mathbf{h}$ as possible. This can be achieved by using a reconstruction loss minimization or maximizing MI between the outputs of GCS and $\mathbf{h}$. We use MI maximization in our implementation. More implementation details can be found in Appendix H.

## 3.3 Analyzing Interpretation Results In Practice

We implement the GCS model in Equation 2 and use attention coefficients in Equation 3 for all relations and entities in the KG for interpretation. Then we analyze these interpretation results to get conclusions for the KI process.

---

[3]Note that the number of graph convolutional layers decides the receptive field of entities. $n$ layers represent that each entity can get information from its $n$-hop neighbors.

As introduced in §3.1, large edge attention coefficients mean that the triples corresponding to the edge are integrated well. To understand CR and CF, we add self-loops on entities in the KG, and use attention coefficients on the self-loops to show how much of the original information is remembered/forgotten for entities. In particular, large self-loop attention coefficients mean that the original entity information is kept well. Thus, we introduce thresholds to analyze our interpretation results as follows[4]. We simply regard triples with attention coefficients $a_{i,j} > 0.1$ on edges as integrated triples. Entities with attention coefficients $0.4 < a_{i,i} < 0.6$ on self-loops as well-learned entities. Entities with $a_{i,i} < 0.1$ on self-loops means CF has happened, where much new factual knowledge information is integrated and original entity information is forgotten. Correspondingly, $a_{i,i} > 0.9$ means CR has happened. For interpretation, attention coefficients on edges are used for triple-wise integration, and those on self-loops are for entity-wise integration.

## 4 EXPERIMENTS

We first introduce two knowledge-enhanced LMs considered in this work: K-Adapter (Wang et al., 2021a) and ERNIE (Zhang et al., 2019). KG is integrated in a triple-wise manner in K-Adapter, and entity-wise manner in ERNIE.

Then, we move on to our experiments. First, we verify our GCS model. We prove that GCS can correctly interpret how much KG information is integrated, as well as which set of entities and triples are integrated. After that, we use GCS to interpret the KI process for K-Adapter and ERNIE. We present the interpretation results and find that both K-Adapter and ERNIE have only integrated few triples, but they integrate many entities. Finally, we use our probe to understand which kinds of knowledge is integrated well in these models. In order to do this, we stratify knowledge in terms of various relation type and find that K-Adapter and ERNIE integrate different kinds of factual knowledge to different extents.

### 4.1 KNOWLEDGE-ENHANCED MODELS

Below, we describe the two knowledge-enhanced LMs considered in this work:

**K-Adapter.** K-Adapter takes RoBERTa (Liu et al., 2019b) as the backbone model and inserts three new layers into RoBERTa to learn new knowledge. The final output is concatenated with the output of RoBERTa[5]. During the integration, parameters of RoBERTa are frozen, only parameters of the newly inserted layers are updated. K-Adapter uses the T-REx-rc (ElSahar et al., 2018) dataset for KI, which has an alignment of natural sentences with knowledge triples in Wikidata. For the KI objective, K-Adapter decides whether certain relations exist or not, and classifies relation labels given the aligned sentence. As knowledge is integrated in newly inserted layers, the model no longer needs to use the T-REx-rc dataset for inference, and it can be finetuned like any other pretrained LMs on downstream tasks.

**ERNIE.** ERNIE integrates factual knowledge into BERT (Devlin et al., 2019) directly without introducing extra parameters. The Wikipedia corpus and Wikidata knowledge triples are selected for integration. As there is no provided alignment between natural sentences and knowledge entities, ERNIE uses TAGME (Ferragina & Scaiella, 2010) to extract entity mentions in sentences and aligns them with corresponding entities in KGs. A new objective is designed for KI in addition to the standard MLM and NSP objectives: alignments in the input text are randomly masked, and the model is asked to select aligned entities from KGs. Different from K-Adapter that stores factual knowledge in newly introduced parameters, when ERNIE finds the aligned entity, its embedding obtained from Bordes et al. (2013) is integrated into the output representations. Thus, during inference, the KG and its embeddings are still required to be fed into ERNIE.

### 4.2 GCS VERIFICATION

We design a set of experiments to verify that GCS can indeed correctly interpret the KI process. We first verify that GCS can correctly interpret how much knowledge is integrated using synthetic

---

[4]Note that users may choose different thresholds. We heuristically set these thresholds for our analysis.
[5]Note that here we only consider factual knowledge, thus, the linguistic Adapter is not used.

experiments (§4.2.1). Then, we verify that GCS can correctly interpret which type of knowledge is integrated based on the KI dataset (§4.2.2) and downstream task performance (§4.2.3).

### 4.2.1 SYNTHETIC EXPERIMENT

**Setting.** We create a synthetic KI scenario where different amounts of knowledge is integrated into the LM. Specifically, we first use DeepWalk (Perozzi et al., 2014) to obtain entity embeddings (i.e., KG embeddings) in the KG used for KI. We regard entity embeddings as the entity representations of knowledge-enhanced LMs, and we add Gaussian noise with different noise ratios on entity embeddings and regard it as the entity representations of vanilla LMs. Then, we simulate the KI processes from vanilla LMs (noisy entity embeddings) to knowledge-enhanced LMs (entity embeddings). Large noise ratio means much information of KG is integrated.

**Baselines.** We select several baselines for comparison. *Representation analysis*: we calculate the similarity (e.g., Cosine, Euclidean) between two connected entities to estimate how much knowledge about the corresponding triple is contained in the LM. Finally, the similarity gap between vanilla LMs and knowledge-enhanced LMs can be used to interpret how much knowledge is integrated. *Linear probe*: we design a linear classifier to do link prediction in the KG based on entity representations. The performance (i.e., AUC score) can be used to show how much knowledge is contained in the LM, and its gap shows how much knowledge is integrated. As for GCS, we use the mean value of self-loop attention coefficients to show how much knowledge is integrated: large values mean that little knowledge is integrated. For the convenience of observation, we report one minus its mean value as the integration score. Implementation details are in Appendix F.

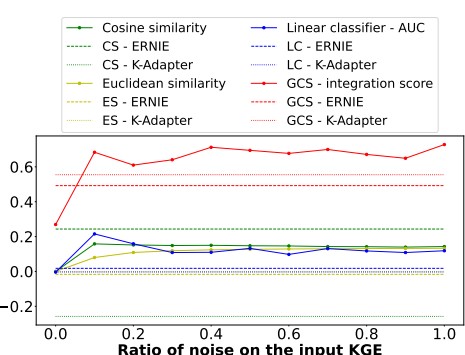

**Figure 3:** Interpretation results of how much knowledge is integrated based on different methods. Solid lines are results of synthetic KI processes. Dotted lines and dashed lines show results of K-Adapter and ERNIE.

**Results.** Figure 3 shows the interpretation results. Solid lines show the results of synthetic KI processes. We find that GCS and representation analysis methods provide correct results: the similarity gap/score increases as the noise ratio increases. However, linear classifier probe can only differentiate whether new knowledge is integrated or not, and fails to tell how much knowledge is integrated (its curve is not very monotone.). In practice, dotted lines and dashed lines show interpretation results of K-Adapter and ERNIE. We find that only GCS provides reasonable results: they integrate little knowledge (equivalent to noise ratio $\approx 5\%$). All baselines fail in interpretation: K-Adapter and ERNIE do not integrate any factual knowledge, or they even forget learned factual knowledge.

### 4.2.2 VERIFICATION USING THE KI DATASET

**Setting.** This experiment is composed of three steps. First, we use GCS to interpret the KI process in K-Adapter and ERNIE, and identify triples and entities that are integrated successfully. Second, we retrain BERT/RoBERTa to get K-Adapter (dropped)/ERNIE (dropped) only using the triples/entities that are identified as successfully integrated. Third, we finetune K-Adapter/ERNIE and their dropped versions on downstream tasks. If GCS correctly interpreted the KI process, the performance of their dropped versions on downstream tasks should be roughly the same as that of K-Adapter/ERNIE.

As introduced in §4.1, K-Adapter and ERNIE integrate knowledge by aligned natural sentences instead of using triples/entities directly. Thus, after we get the interpretation results, i.e., triples with edge attention coefficients larger than $0.1$ or entities with self-loop attention coefficients smaller than $0.9$, we only keep data aligned with the integrated knowledge. Specifically, we drop sentences for K-Adapter and entity embeddings (obtained by Bordes et al. (2013)) for ERNIE. Then, we finetune them on two downstream tasks (entity typing) that K-Adapter and ERNIE outperform RoBERTa and BERT most significantly: OpenEntity (Choi et al., 2018) and FIGER (Ling et al., 2015). Implementation details of reproduction and GCS can be found in Appendix G and Appendix H.

**Results.** We only keep 10.09% natural sentences that aligned with successfully integrated triples for K-Adapter (dropped), and 61.72% entity embeddings (obtained by Bordes et al. (2013)) for ERNIE (dropped). More details

**Table 1:** Performance of K-Adapter, ERNIE, and their dropped versions on entity typing on the OpenEntity and FIGER datasets.

| Model | OpenEntity | | | FIGER | | |
|---|---|---|---|---|---|---|
| | P | R | F1-Micro | P | R | F1-Micro |
| RoBERTa | 76,98 | 73.42 | 75.16 | 65.26 | 88.72 | 75.20 |
| K-Adapter | 76.63 | 75.26 | 75.94 | 67.50 | 88.79 | 76.69 |
| K-Adapter (dropped) | 75.95 ↓ | 75.95 ↑ | 75.95 ↑ | 67.29 ↓ | 88.88 ↑ | 76.59 ↓ (6.71%) |
| BERT | 79.68 | 65.70 | 72.02 | 75.59 | 62.32 | 68.32 |
| ERNIE | 78.24 | 68.75 | 73.19 | 77.39 | 65.81 | 71.13 |
| ERNIE (dropped) | 78.11 ↓ | 71.43 ↑ | 74.62 ↑ | 77.38 ↓ | 64.90 ↓ | 70.60 ↓ (18.86%) |

are in Appendix I. From Table 1, we can find that even if we drop large amount of KI data in this way, the performance of K-Adapter (dropped) and ERNIE (dropped) on entity typing task is roughly the same as original (reproduced) versions, and they are obviously better than that of BERT and RoBERTa. We further verify GCS by introducing a random dropping strategy as comparison on the OpenEntity dataset. Detailed results can be found in Appendix J.

### 4.2.3 VERIFICATION WITH A DOWNSTREAM TASK

**Settings.** In this experiment, we combine the interpretation results of GCS with a downstream task. Specifically, we align entities in the KI dataset and the OpenEntity dataset based on their *Wikidata Q identifier*[6]. For the entity typing task (OpenEntity dataset), we drop the finetuning test data samples that aligs with the integrated knowledge and non-integrated entities (called *drop-IE* test set and *drop-UE* test set), and test K-Adapter and ERNIE on the two dropped test sets. If GCS correctly interpret the KI process, knowledge-enhanced LMs should perform better on the *drop-UE* test set and worse on the *drop-IE* test set.

**Results.** Table 2 presents detailed results. We can find that for K-Adapter, the gap is not very obvious. We hypothesize that this may be because of the differences in the finetuning objective and the KI objective, and because the knowledge integrated in K-Adapter may change during finetuning. As for ERNIE, the gaps are signifi-

**Table 2:** Performance change of K-Adapter and ERNIE on the OpenEntity dataset with different test sets.

| Model (Test set) | OpenEntity | | | |
|---|---|---|---|---|
| | Left test set | P | R | F1-Micro |
| K-Adapter (drop-IE) | 37.44% | − 0.33 | − 0.37 | − 0.35 |
| K-Adapter (drop-UE) | 64.46% | − 0.18 | + 1.12 | + 0.47 |
| ERNIE (drop-IE) | 27.28% | − 18.20 | − 25.14 | − 22.67 |
| ERNIE (drop-UE) | 66.87% | − 0.31 | + 3.08 | + 1.57 |

cant. The performance (F1-Micro) on the test set (drop-IE) is 20 F1 points worse than that on the complete test set. These results also verify that GCS can correctly interpret which set of knowledge is integrated.

### 4.3 GCS FINDINGS

After verifying GCS with three groups of experiments, we analyze the interpretation results. From a macro view, we find that both K-Adapter and ERNIE integrate few knowledge triples ($\approx 20\% - 30\%$) and some knowledge entities ($\approx 60\% - 70\%$). Detailed results can be found in Appendix K. From a micro view, we classify knowledge based on relation types (in terms of their topology type and Wiki data type) and analyze how K-Adapter and ERNIE integrate them respectively.

**KI analysis for K-Adapter and ERNIE in terms of relation topology.** We classify relations into three types based on their topology features. Specifically, relations that connect two leaf nodes (entities) in the KG are $1 - 1$ relations, and relations that connect two center nodes (entities) in the KG are $N - M$ relations. Others are $N - 1$ relations. For the analysis results in terms of different types of relations, we report the percentage of successfully integrated triples and entities for K-Adapter and ERNIE. Besides, we also present the percentage of catastrophic remembered (CR) entities and catastrophic forgotten entities (CF).

Table 3 presents specific results. We find that for K-Adapter, triples with $N - M$ relations are not captured well. However, K-Adapter integrates triples with $1 - 1$ relations well. This phenomenon is common for Transformer encoders, where knowledge with complex structure cannot be captured well (Petroni et al., 2019). ERNIE shows different behaviors. We find that entities connected with $1 - 1$ and $N - M$ relations are captured well. But for entities connected to $N - 1$ relations, they

---

[6]https://www.wikidata.org/wiki/Q43649390

**Table 3:** Analysis of KI interpretation results for K-Adapter and ERNIE in terms of different types of relations (topology feature). The percentages of integrated triples/entities, as well as of CR and CF entities for each type of relations are presented.

| Model / Statistics | K-Adapter (Wang et al., 2021a) on T-REx-rc | | | |
|---|---|---|---|---|
| | $1 - 1$ relation | $N - 1$ relation | $N - M$ relation | Total |
| # of triples | 21,690 | 813,674 | 1,729,644 | 2,565,008 |
| Integrated triple percentage | 58.89% | 38.39% | 24.00% | 28.86% |
| # of connected entities | 21,690 | 406,837 | 352,748 | 781,275 |
| CR entity percentage | 41.11% | 31.72% | 26.02% | 29.41% |
| CF entity percentage | 26.40% | 30.29% | 40.89% | 34.97% |
| Model / Statistics | ERNIE (Zhang et al., 2019) on Wikidata | | | |
| | $1 - 1$ relation | $N - 1$ relation | $N - M$ relation | Total |
| # of connected entities | 1,799 | 529,186 | 2,744,549 | 3,275,534 |
| Integrated entity percentage | 70.65% | 42.86% | 73.33% | 68.39% |
| CR entity percentage | 29.41% | 56.07% | 26.67% | 38.28% |
| CF entity percentage | 23.18% | 8.65% | 37.10% | 32.49% |

are not integrated well. Since ERNIE relies on KG embedding to learn structure knowledge, KI is highly consistent with the quality of the KG embedding provided in Bordes et al. (2013). Regarding CR and CF, we find that for both K-Adapter and ERNIE, CR happens more often to entities in simple structures (i.e., connected to $1 - 1$ or $1 - N$ relations), while CF is more common for entities in complex structures (i.e., connected to $N - M$ relations).

**KI analysis for K-Adapter and ERNIE in terms of relation's Wiki features.** We select six relations aligned with roughly the same number of sentences in the T-REx-rc dataset (see Appendix L for statistics) and categorize them into three groups based on the *Wiki Count* and *Wiki data type*[7]: low-frequency (LF) relations, time-related (TR) relations, and high-frequency (HF)

**Table 4:** The interpretation of KI for K-Adapter in terms of relations. We list 6 relations and classify them into three types based on the Wiki Count and Wiki data type. The ratio of integrated knowledge triples are reported.

| Relation label | T-REx-rc | | |
|---|---|---|---|
| | Wiki Count | Wiki data type | Integrated triple percentage |
| Place of birth (LF) | 2,850,424 | Wikibase item | 10.95% |
| Part of (LF) | 4,164,470 | Wikibase item | 17.25% |
| Date of death (TR) | 2,637,358 | Time | <0.01% |
| Date of birth (TR) | 5,294,649 | Time | <0.01% |
| Located in the administrative territorial entity (HF) | 10,776,120 | Wikibase item | 6.13% |
| Country (HF) | 14,174,811 | Wikibase item | 0.12% |
| Total | - | - | 10.09% |

relations. From Table 4, we can find that even if LF relations has roughly the same *Wiki Count* as TR relations. However, since the *Wiki data type* of the latter set is "Time", triples with those relations cannot be integrated by K-Adapter. We speculate that this is because Transformer encoders do not capture information about time well (Dhingra et al., 2021; Zhou et al., 2021). When comparing LF relations and HF relations, we find that if relations have small Wiki Count, knowledge triples are easier to be captured during KI.

Above experiments show that interpretation results of GCS are consistent with existing analysis works, which also verify GCS indirectly. Besides, we design a case study experiment in Appendix M, where we find that CR often happens to rare entities (small Google Ngrams), while CF often happens to popular entities (large Google Ngrams). Moreover, we study the correlation between the edge attention coefficient and the number of aligned sentences for K-Adapter in Appendix N, and find that the *Pearson Correlation Coefficient* is -0.0055. It implies that simply increasing the KI dataset may not help LMs integrate unlearn knowledge.

## 5 LIMITATIONS AND FUTURE WORK

In this paper, we illustrate that the graph attention can be used to interpret the KI process for knowledge-enhanced LMs, and thus propose a Graph Convolutional Simulator (GCS) that is capable of correctly interpreting existing knowledge-enhanced LMs. In our experiments, we verify GCS and use it to obtain interesting findings. There are some limitations of our work. We simplify the KG without considering edge direction, labels, multi-edges, entity descriptions, and timestamps. These can be considered by future works. Besides, GCS only provides a way to interpret the knowledge integration. Once we have an understanding of the KI process, improving the integration quality still remains challenging.

---

[7]https://www.wikidata.org/wiki/Wikidata:Database_reports/List_of_properties/all

## REPRODUCIBILITY STATEMENT

We will publish the code, as well as the interpretation results after the review process. The design of GCS can be found in §3. The implementation details about the KI for K-Adapter and ERNIE can be found in Appendix G. And details about GCS can be found in Appendix H.

## ETHICS STATEMENT

While our probe models are not tuned for any specific real-world application, our methods could be used in sensitive contexts such as legal or health-care settings; and it is essential that any work that builds on our approaches undertake extensive quality-assurance and robustness testing before using it in their setting.

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

# A  NOTATIONS

**Table 5:** Notations and their descriptions

| Notation | Description |
|---|---|
| $\mathcal{G}$ | The knowledge graph for KI |
| $\mathcal{V}$ | The set of entities/nodes of KG |
| $\mathcal{E}$ | The set of relations/edges of KG |
| $v_i$ | The entity/node indexed as $i$ in the KG |
| $t_i$ | The entity text attached on $v_i$ |
| $\text{LM}(\cdot)$ | The language model, where the input is entity text, and the output is its representation |
| $\mathcal{N}_{v_i}$ | The set of neighbors (entities/nodes) connected to $v_i$ |
| $\mathcal{G}(v_i)$ | The local graph structure in terms of $v_i$ |
| $\mathbf{x}$ | The random variable of the entity representation |
| $\boldsymbol{x}_i$ | The entity representations of $v_i$ |
| $\mathbf{g}$ | The random variable of the local graph structure |
| $\text{MI}(\cdot;\cdot)$ | The mutual information between two random variables |
| $\boldsymbol{A}$ | The adjacency matrix of KG |
| $|\mathcal{V}|$ | The number of entities/nodes in KG |
| $\mathbb{R}$ | The set of real numbers |
| $\boldsymbol{I}$ | The identity matrix |
| $\boldsymbol{D}$ | The degree matrix of KG |
| $\boldsymbol{L}_n$ | The normalized Laplacian matrix |
| $\text{diag}(\cdot)$ | The diagonalization operation |
| $\boldsymbol{U}$ | The matrix of eigenvectors |
| $\boldsymbol{\Lambda}$ | The diagonal matrix of eigenvalues |
| $\lambda_i$ | The $i$-th eigenvalue |
| $\boldsymbol{X}$ | The set of entity representations in terms of $\mathcal{V}$ |
| $C$ | The dimension of entity representations; The number of channels |
| $\text{GFT}(\cdot)$ | The graph Fourier transformation |
| $\text{RGFT}(\cdot)$ | The inverse graph Fourier transformation |
| $g_\Theta$ | The graph filter parameterized by parameter $\Theta$ |
| $\boldsymbol{H}$ | The entity representations given by a knowledge-enhanced LM |
| $\mathbf{h}$ | The random variable of the entity representation given by a knowledge-enhanced LM |
| $f(\cdot)$ | The mapping that can transform $\mathbf{x}$ to $\mathbf{h}$ |
| $\epsilon$ | The error of the approximation |
| $\text{sigmoid}(\boldsymbol{x})$ | The Sigmoid function $\text{sigmoid}(\cdot) = \frac{1}{1+e^{-\boldsymbol{x}}}$ |
| $n$ | The number of layers of the neural network for apporximation |
| $\boldsymbol{W}$ | The weight matrix |
| $\boldsymbol{x}$ | The input vector |
| $\boldsymbol{b}$ | The bias |
| $\lambda'_0$ | The minimum eigenvalue of the weight matrix $\boldsymbol{W}$ |
| $\text{MLP}_n(\cdot)$ | The bijective MLP function |
| $\text{GC}(\cdot,\cdot)$ | The graph convolution function |
| $\text{GCS}_{\theta_1}$ | The GCS model parameterized by $\theta_1$ |
| $\mathcal{L}$ | The objective of the optimization |
| $\boldsymbol{Z}$ | The output of GCS, i.e., set of output entity representations |
| $\mathbf{z}$ | The random variable of the output of GCS |
| $\sup$ | The supremum value |
| $T$ | A class of functions |
| $\mathcal{F}$ | Any class of functions |
| $\Omega$ | The domain of a function |
| $T_{\theta_2}$ | A class of functions parameterized by $\theta_2$, i.e., neural networks |
| $\mathbb{P}$ | The probability distribution |
| $\mathbb{P}^{|\mathcal{V}|}$ | The empirical distribution with $|\mathcal{V}|$ samples |
| $\text{NN}_\sigma(\cdot|\theta')$ | The neural network with activation function $\sigma(\cdot)$ and parameterized by $\theta'$ |
| $|\boldsymbol{U}|$ | The norm of matrix $\boldsymbol{U}$ |
| $\boldsymbol{A}_n$ | The normalized adjacency matrix |
| $\hat{\boldsymbol{X}}$ | The ground-truth entity representations/node features |
| $\hat{\boldsymbol{X}}^*$ | The variable matrix |
| $\mathbf{Tr}(\cdot)$ | The trace of a matrix |
| $\epsilon_1, \epsilon_2$ | The error bound of entity representations/node features and adjacency matrix |
| $\gamma$ | The Lagrangian multiplier |
| $p(t)$ | The characteristic polynomial for weight matrix $\boldsymbol{W}$ |
| $\det(\cdot)$ | The determinant of a matrix |

# B   PROOF OF THEOREM 2

*Proof.* As aforementioned, the graph Fourier transformation $\text{GFT}(\cdot)$ and its inverse transformation $\text{RGFT}(\cdot)$ in terms of the KG $\mathcal{G}$ can be written as

$$\text{GFT}(\boldsymbol{X}) = \boldsymbol{U}^T \boldsymbol{X}$$
$$\text{RGFT}(\text{GFT}(\boldsymbol{X})) = \boldsymbol{U}\text{GFT}(\boldsymbol{X}) = \boldsymbol{U}\boldsymbol{U}^T \boldsymbol{X} = \boldsymbol{X}.$$

The second equation can be derived since $\boldsymbol{U}$ is the set of eigenvectors of the normalized Laplacian matrix in terms of $\mathcal{G}$, which is orthogonal.

According to the universal approximation theorem (Cybenko, 1992), in general, we can use one-layer neural networks (arbitrary width) with the sigmoid activation function to fit any functions. Ohn & Kim (2019) bound the approximation with both the width and depth, and supports more activation functions. Based on the conclusion of Ohn & Kim (2019), we know that given a mapping $f'(\cdot)$, for any $\epsilon' > 0$, there exists a neural network parameterized by $\theta'$ s.t.

$$|f'(\cdot) - \text{NN}_\sigma(\cdot|\theta')| < \epsilon'.$$

Note that there are some constraints about the input and the model architecture, i.e., width and depth. We leave out those details for simplicity. More details can be found in Ohn & Kim (2019).

Since $\mathbf{h}$ is obtained by integrating $\mathbf{g}$ into $\mathbf{x}$, we can simplify the mapping in the graph space by researching on the transformation from $\text{GFT}(\mathbf{x})$ to $\text{GFT}(\mathbf{h})$[8]. Assume the mapping satisfies $f'(\text{GFT}(\mathbf{x})) = \text{GFT}(\mathbf{h})$. Then we have

$$|f'(\text{GFT}(\mathbf{x})) - \text{NN}_\sigma(\text{GFT}(\mathbf{x})|\theta')| < \epsilon'.$$

Consider that we have $f(\mathbf{x}, \mathbf{g}) = \mathbf{h} = \text{RGFT}(f'(\text{GFT}(\mathbf{x})))$. If we assign $\epsilon' = \frac{\epsilon}{|\boldsymbol{U}|} > 0$, we have

$$|\boldsymbol{U}| \cdot |f'(\text{GFT}(\mathbf{x})) - \text{NN}_\sigma(\text{GFT}(\mathbf{x})|\theta')| < \epsilon.$$

Since we know that

$$\boldsymbol{U} \cdot f'(\text{GFT}(\mathbf{x})) = \text{RGFT}(f'(\text{GFT}(\mathbf{x}))) = \mathbf{h} = f(\mathbf{x}, \mathbf{g}),$$

we have

$$|f(\mathbf{x}, \mathbf{g}) - \text{RGFT}(\text{NN}(\text{GFT}(\mathbf{x})))| < |\boldsymbol{U}| \cdot |f'(\text{GFT}(\mathbf{x})) - \text{NN}_\sigma(\text{GFT}(\mathbf{x})|\theta')| < \epsilon,$$

where $\text{NN}(\cdot)$ is parameterized by $\theta'$ with activation function $\sigma$ as $\text{NN}_\sigma(\cdot|\theta')$. And without loss of generality, we assume it is composed of $n$ layers. □

# C   GRAPH CONVOLUTIONS FOR KI SIMULATION (PROOF)

*Proof.* The basic idea of this proof can be found in Figure 2. We first simply prove that the graph Fourier transformation is bijective. Similarly, the nonlinear activation function can be proved bijective. Then, we show that information gain and loss can only happen in the linear function in graph space. After that, we briefly illustrate that linear function in graph space is graph convolution operation. Finally, we prove that graph attention works as edge denoising, and we can use it to interpret the KI.

---

[8]In next proof, we illustrate that the linear transformation in the graph space is graph convolution, which integrates the graph information into entities. Thus, in the graph space, we do not need to regard $\mathbf{g}$ as an input. More formally description can be found in Chen et al. (2019); Keriven & Peyré (2019).

## C.1 STEP 1

GFT($\cdot$), RGFT($\cdot$), **and** sigmoid($\cdot$) **are bijective.** Given two entity representations $\boldsymbol{x}_i$, $\boldsymbol{x}_j$ and the matrix of eigenvectors of the KG as $\boldsymbol{U}$, suppose that GFT($\boldsymbol{x}_i$) = GFT($\boldsymbol{x}_j$). Then, we have

$$\boldsymbol{U}^T \boldsymbol{x}_i = \boldsymbol{U}^T \boldsymbol{x}_j.$$

Since $\boldsymbol{U}^T$ are set of eigenvectors and are by definition nonzero, we have

$$\boldsymbol{x}_i = \boldsymbol{x}_j.$$

If $\boldsymbol{x}_i = \boldsymbol{x}_j$, it is easy to get GFT($\boldsymbol{x}_i$) = GFT($\boldsymbol{x}_j$). Thus, graph Fourier transformation is bijective.

As for the nonlinear activation function, since we consider neural networks composed of MLP layers, the activation function is sigmoid($\cdot$) function. It is easy to find that its inverse function is $f(\boldsymbol{y}) = \ln(1 - \frac{1}{\boldsymbol{y}})$. Similarly, we can prove that it is bijective as well.

## C.2 STEP 2

**Information gain and loss can only happen in the linear function in graph space.** Based on the invariance of MI (Kraskov et al., 2004), we have

$$\begin{aligned}
\mathrm{MI}(\mathbf{x}, \mathbf{g}) &= \mathrm{MI}(\mathrm{GFT}(\mathbf{x}), \mathbf{g}), \\
\mathrm{MI}(\mathbf{x}, \mathbf{g}) &= \mathrm{MI}(\mathrm{RGFT}(\mathbf{x}), \mathbf{g}), \\
\mathrm{MI}(\mathbf{x}, \mathbf{g}) &= \mathrm{MI}(\mathrm{sigmoid}(\mathbf{x}), \mathbf{g}).
\end{aligned} \tag{4}$$

Since we know that

$$\mathrm{MI}(\mathbf{h}, \mathbf{g}) - \mathrm{MI}(\mathbf{x}, \mathbf{g}) > 0,$$

and the neural network can well approximate the mapping, we have

$$\begin{aligned}
\mathrm{MI}(\mathbf{h}, \mathbf{g}) - \mathrm{MI}(\mathbf{x}, \mathbf{g}) &\approx \mathrm{MI}(\mathrm{RGFT}(\mathrm{NN}(\mathrm{GFT}(\mathbf{x}))), \mathbf{g}) - \mathrm{MI}(\mathbf{x}, \mathbf{g}) \\
&= \mathrm{MI}(\mathrm{NN}(\mathrm{GFT}(\mathbf{x})), \mathbf{g}) - \mathrm{MI}(\mathrm{GFT}(\mathbf{x}), \mathbf{g}) > 0.
\end{aligned}$$

If we write NN($\cdot$) with $n$ MLP layers as $n \times \sigma(\mathrm{Linear}(\cdot))$, we have

$$\mathrm{MI}(n \times \sigma(\mathrm{Linear}(\mathrm{GFT}(\mathbf{x})), \mathbf{g}) - \mathrm{MI}(\mathrm{GFT}(\mathbf{x}), \mathbf{g}) > 0.$$

Recursively with equations 4, it is easy to get that MI only changes in the $\mathrm{Linear}(\cdot)$ functions.

## C.3 STEP 3

**The linear function in the KG space (i.e., graph spectral domain) is the graph convolution operation.** Even if many existing works (Sandryhaila & Moura, 2014; Bruna et al., 2014; Kipf & Welling, 2017) have provided clear descriptions, we simply re-illustrate it under the multi-channel setting. Consider the graph filter in Bruna et al. (2014) as an exmaple.

For a linear function $f(\boldsymbol{x}) = \boldsymbol{W} \times \boldsymbol{x}$, its weight matrix $\boldsymbol{W} \in \mathbb{R}^{F \times C}$ is parameterized by $\Theta \in \mathbb{R}^{F \times C}$. If the parameters are not shared for all nodes, the input $\boldsymbol{X} \in \mathbb{R}^{|\mathcal{V}| \times C}$ can be rescaled in $\mathbb{R}^{|\mathcal{V}| \times C \times 1}$, and the weight matrix is $\boldsymbol{W} \in \mathbb{R}^{|\mathcal{V}| \times F \times C}$ parameterized by $\Theta \in \mathbb{R}^{F \times C \times |\mathcal{V}|}$. The output of this linear function is mapped in $\mathbb{R}^{|\mathcal{V}| \times F}$.

Consider the signal in graph convolution, i.e., all $\boldsymbol{x}$ in $\boldsymbol{X} \in \mathbb{R}^{|\mathcal{V}| \times C}$. Since parameters are not shared (Bruna et al., 2014), for one graph filter, the parameters in $g_\Theta$ is in $\mathbb{R}^{C \times |\mathcal{V}| \times |\mathcal{V}|}$ that is parameterized by $\Theta \in \mathbb{R}^{C \times |\mathcal{V}|}$ with simple diagonalization. If we have $F$ different graph filters for the convolution, $g_\Theta$ is in $\mathbb{R}^{F \times C \times |\mathcal{V}| \times |\mathcal{V}|}$ that is parameterized by $\Theta \in \mathbb{R}^{F \times C \times |\mathcal{V}|}$. Here, the graph Fourier transformation of $\boldsymbol{X}$ is GFT($\boldsymbol{X}$) $\in \mathbb{R}^{|\mathcal{V}| \times C}$, which can be rescaled in $\mathbb{R}^{1 \times |\mathcal{V}| \times C \times 1}$ with simple diagonalization. The output is in $\mathbb{R}^{F \times |\mathcal{V}| \times |\mathcal{V}| \times 1}$. Note that since the parameters in the graph filter is diagonalized, we can rescale the output in $\mathbb{R}^{|\mathcal{V}| \times F}$.

If we regard the weight matrix $\boldsymbol{W}$ as the parameters in the graph filter $g_\Theta$, the input matrix $\boldsymbol{X}$ as the signal, obviously, the linear function in the graph space is the graph convolution operation. $\qquad \square$

## D   GRAPH ATTENTION FOR KI INTERPRETATION (PROOF)

*Proof.* **Graph attention works as edge denoising[9].** Graph attention works as a better graph convolution filter, since it can adaptively learn the optimal convolution weights (i.e., attention coefficients). Consider a graph signal denoising problem that we aim to extract the ground-truth node features $\hat{X}$ and edge weights $\hat{A}_n$ from a graph $\mathcal{G} = (\mathcal{V}, \mathcal{E}, A_n)$ with noise in both node features $X$ and edge weights $A_n$. Here, $A_n$ is the normalized adjacency matrix $A_n = D^{-1/2} A D^{-1/2}$. To this end, we formulate the optimization problem under the assumption that the ground-truth node features $\hat{X}$ are smooth w.r.t the ground-truth adjacency matrix $\hat{A}_n$ and the noise in the graph can be upper-bounded:

$$
\begin{aligned}
\hat{X}^*, \hat{A}_n^* = \underset{\hat{X}, \hat{A}_n}{\arg\min} \mathrm{Tr}\left(\hat{X}\hat{L}_n^T \hat{X}\right) \\
\text{s.t. } \|\hat{X} - X\|_2^2 \le \epsilon_1, \\
\|\hat{A}_n - A_n\|_2^2 \le \epsilon_2,
\end{aligned}
\tag{5}
$$

where $\hat{L} = I - \hat{A}$, $\epsilon_1, \epsilon_2 \in \mathbb{R}$, are the level of noise in node features and edge weights, respectively. $\mathrm{Tr}(\cdot)$ indicates the trace of a matrix. By Lagrange multipliers methods, we can obtain the solution as following:

$$
\hat{X}^* = \frac{\gamma}{1+\gamma}\left(I - \frac{1}{1+\gamma}\hat{A}_n^*\right),
\tag{6}
$$

$$
\hat{A}_n^* = A_n + \sqrt{\epsilon_2}\frac{\hat{X}^* \hat{X}^{*\top}}{\|\hat{X}\|_2^2},
\tag{7}
$$

where $\gamma > 0$ is the Lagrangian multiplier. Note that the attention coefficients of GAT (Velickovic et al., 2018) and AGNN (Thekumparampil et al., 2018) are obtained by (without less of generality, we show the results in the first-layer) equation 8 and equation 9, respectively:

$$
a_{i,j} = \mathrm{softmax}\left(\mathrm{leakyReLU}\left(\mathbf{a}^\top\left[WX_i\|WX_j\right]\right)_{j \in \mathcal{N}_i \cup \{i\}}\right),
\tag{8}
$$

$$
a_{i,j} = \mathrm{softmax}\left(\left[\beta\frac{H_i^\top H_j}{\|H_i\|\|H_j\|}\right]_{j \in \mathcal{N}_i \cup \{i\}}\right),
\tag{9}
$$

where $H = \mathrm{ReLU}(XW)$, $\mathbf{a}$, $W$ in equation 8, and $\beta$, $W$ in equation 9 are learnable parameters. The attention coefficents of GAT and AGNN are then used as the weights of aggregating the neighbohood information of nodes. As we can see that equation 7, equation 8, and equation 9 are in a form of measuring the similarity between paired node features. Similar to the denoised edge weights obtained in equation 7, the attention coefficents (i.e. the aggregation weights) between a node and its neighborhoods are proportional to the similarity of their node embeddings. Therefore, the attention coefficients of GAT and AGNN can be regarded as the results of denoised weights on the existing edges in a graph, i.e., the graph attentions are implicitly denoising the edge weights.

In general case, graph attention functions as denoising edge weights. The input is noisy representations and the output is the groundtruth. Attention coefficients show how much distortion is corrected during the convolution operation. For example, if the input representations are also groundtruth, there is no need to fetch information from neighbors to get output. And edge weights will be reduced to 0, i.e., attention coefficients on edges are calculated as 0. If the input representations are very noisy, i.e., much noise are removed, attention coefficients on edges should be large to restore the groundtruth signal. Therefore, in the KI scenario, we can use attention coefficients in graph attention in graph convolution layer to interpret the KI process such as how much triple information is integrated. As for the CR and CF, equally, we can use the attention coefficients on the self-loop edges for interpretation, such as how much original information is remembered/forgotten.

$\square$

---

[9]The detailed proof can be found in Fu et al. (2020).

## E   MLP CAN BE BIJECTIVE (PROOF)

**Theorem 5.** *Give an MLP layer denoted as* $\mathrm{MLP}(\boldsymbol{x}) = \mathrm{sigmoid}(\boldsymbol{W}\boldsymbol{x} + \boldsymbol{b})$. *If* $\boldsymbol{W}$ *is a square matrix, there exist a constant* $\lambda_0' > 0$ *that for any* $0 < \epsilon < \lambda_0'$, *the function below is bijective:*

$$\mathrm{MLP}_n(\boldsymbol{x}) = \mathrm{sigmoid}((\boldsymbol{W} - \epsilon\boldsymbol{I})\boldsymbol{x} + \boldsymbol{b}). \tag{10}$$

*Proof.* We first prove that two bijective function compositions are still bijective. Then, we prove that adding a small noise on MLP weight matrix can make it bijective.

Give two function $f_1(\cdot)$ and $f_2(\cdot)$. Suppose they are injective and suppose $f_1(f_2(\boldsymbol{x})) = f_1(f_2(\boldsymbol{y}))$. Since we know that $f_1(\cdot)$ is injective, we have $f_2(\boldsymbol{x}) = f_2(\boldsymbol{y})$. Similarly, since $f_2(\cdot)$ is injective, we have $\boldsymbol{x} = \boldsymbol{y}$. Thus $f_1(f_2(\cdot))$ is injective. Suppose $f_1(\cdot)$ and $f_2(\cdot)$ are surjective and $\boldsymbol{z} \in C$. Since we know that $f_1(\cdot)$ is surjective, there exists a set of $\boldsymbol{y} \in B$ with $f_1(\boldsymbol{y}) = \boldsymbol{z}$. Similarly, since $f_2(\cdot)$ is surjective, there exists a set of $\boldsymbol{x} \in A$ with $f_2(\boldsymbol{x}) = \boldsymbol{y}$. Then, we have $\boldsymbol{z} = f_1(f_2(\boldsymbol{x}))$ and so $\boldsymbol{z}$ is onto $f_1(f_2(\cdot))$. Thus, $f_1(f_2(\cdot))$ is surjective. Therefore, if $f_1(\cdot)$ and $f_2(\cdot)$ are bijective, $f_1(f_2(\cdot))$ is also bijective.

To prove that the special MLP is bijective, consider an MLP function as

$$\mathrm{MLP}(\boldsymbol{x}) = \sigma(\boldsymbol{W}\boldsymbol{x} + \boldsymbol{b}),$$

where $\boldsymbol{W} \in \mathbb{R}^{C \times C}$ is the weight matrix and $\boldsymbol{b} \in \mathbb{R}^C$ is the bias. Let

$$p(t) = \prod_{i=1}^{C}(\lambda_i' - t)$$

be the characteristic polynomial for weight matrix $\boldsymbol{W}$. Here $\lambda_i'$ are eigenvalues of matrix $\boldsymbol{W}$. Without loss of generality, let $|\lambda_0'| = \min_i |\lambda_i'|$. Then, we know that for any constant $0 < \epsilon < |\lambda_0'|$, we have

$$\det(\boldsymbol{W} - \epsilon\boldsymbol{I}) = p(\epsilon) \neq 0.$$

Thus, if the perturbation $\epsilon$ is small enough, the perturbed matrix $\boldsymbol{W}' = \boldsymbol{W} - \epsilon\boldsymbol{I}$ is nonsingular. Consider the fact that the nonlinear activation function $\sigma(\cdot)$ is $\mathrm{sigmoid}(\cdot)$ function, which is bijective. Therefore, the special MLP function $\mathrm{MLP}_n(\cdot)$ is bijective. And there is no information loss.   $\square$

## F   IMPLEMENTATION DETAILS OF THE SIMULATION EXPERIMENT

**DeepWalk.** We implement DeepWalk to get KG embeddings. The hyperparameters are set as the same as its default values[10]: the number of walks is set as 10, and the walk length is set as 40. The dimension of embeddings is set as 128. Note that since knowledge triples only contain neighbors within one-hop, we select the window size as 1.

**Noisy KG embeddings**. We add Gaussian noisy on the KG embeddings with different ratios. For example, if we add 10% ratio of noise, it can be written as

$$10\% \text{ noisy KGE} = 0.9 * \text{KGE} + 0.1 * \text{noise}.$$

**Similarity**. The Cosine similarity is implemented as

$$1 - \text{Cosine distance}(,),$$

and the Euclidean similarity is implemented as

$$1 - \text{Euclidean distance}(,).$$

**Linear probe**. We design a linear classifier to do link prediction for the KG. Note that we cannot implement the linear classifier on the large and sparse KG to do link prediction directly. To get meaningful results, we do negative sampling with sample number as 5. Then, we select to report the AUC (*Area Under the Curve*) score since other metrics cannot be differentiable under this condition.

---

[10]https://github.com/phanein/deepwalk

We randomly split the edge set of the KG into two even sets for training and test. To reduce variance, we repeat the experiment for 10 times.

**GCS**. We train our GCS model with input as noisy KGE and output as KGE. Note that here noisy KGE and KGE are in the same space, and the task is simpler compared to practical KI interpretation. Thus, we set the epoch number as 10, and use the reconstruction loss (i.e., mean absolute error) for efficiency. Other hyperparameters are set as the same as introduced in Appendix H.

# G   IMPLEMENTATION DETAILS OF LMs

**KI.** To ensure that the experiment settings are fair, we set hyperparameters as the default values. For **K-Adapter**, the code and hyperparameters for KI that we use are from the official projects[11] published by the authors (Wang et al., 2021a). The only two differences are that: we use PyTorch *float 32* instead of *float 16* since BERT and RoBERTa that we use are *float32*, and we use 4 NVIDIA Tesla V100 GPUs for KI training. For **ERNIE**, things are the same. All hyperparameters for KI are set as their default values[12]. Similarly, *float 16* of PyTorch is changed to *float 32*, and we do the integration with 4 NVIDIA Tesla V100 GPUs. Note that the dataset that ERNIE used for KI is Wikipedia, since the code is to fetch latest version of it, the data that we use could be slightly different. Therefore, for both ERNIE and K-Adapter, to ensure the fairness, we reproduce their KI, and report the results of reproduced models instead of results provided in their papers.

**Finetuning.** As for the downstream tasks, all the hyperparameters are consistent with the official project: either they are given in the project or in the README. In the same way, *float 32* and 4 NVIDIA Tesla V100 GPUs are chosen to make sure that the comparison is fair. Note that for K-Adapter and ERNIE, the best performance for different datasets is achieved in different settings. For example, the best performance for K-Adapter on the OpenEntity dataset is achieved with single GPU, but on the TACRED dataset is achieved with four GPUs. Since we focus on the relative performance instead of the best one, we run finetuning on 4 NVIDIA Tesla V100 GPUs for all downstream tasks and all LMs (as well as BERT and RoBERTa).

**Table 6:** Statistics of T-REx-rc and Wikidata

| Statistics
Datasets | # of entities | # of triples | # of aligned sentences | # of entities (optimization) | # of triples (optimization) |
|---|---|---|---|---|---|
| T-REx-rc | 781,275 | 1,282,504 | 5,565,478 | - | - |
| Wikidata | 3,275,534 | 12,849,311 | - | 1,344,393 | 3,240,272 |

The datasets that K-Adapter and ERNIE use are T-REx-rc and Wikidata, some statistics of them are given in Table 6.

# H   IMPLEMENTATION DETAILS OF GCS

In this section, we introduce details of implementing GCS. In practice, GCS is composed of 3 layers: bijective MLP layer, graph convolutional layer, and another bijective MLP layer. As for bijective MLP layers, since weight matrices in them are square matrices, the dimension would remain unchanged: 1024 for K-Adapter and 768 for ERNIE. The nonlinear activation functions are set as $\text{ELU}(\cdot)$ function, which is also bijective. The learning rate is set as $1e^{-3}$, and the dropout rate of the first two MLP layers is $0.2$.

Regarding the graph attention, to make sure interpretation results are stable, we apply multi-head attention mechanism, where the number of attention head is set as $8$. Entity representations are first embedded into a space with the dimension as $64$. Then, the embedded representations are used to calculate the attention coefficients. Note that since the purpose is to interpret and analyze the KI process, we do not split datasets for KI. Considering that GCS model is very simple for large KGs, overfitting is unlikely to happen. Thus, we optimize GCS for the whole datasets. Specifically, for K-Adapter, the whole KG is used for optimization, and results are used for interpretation. And

---

[11]https://github.com/microsoft/K-Adapter
[12]https://github.com/thunlp/ERNIE

for ERNIE, since the KG is very large, we sample a small subgraph with $1,344,393$ entities and $3,240,272$ triples for optimization (see Table 6), and then implement the optimized GCS on the whole KG for interpretation.

The objective function of training GCS can be reconstruction loss minimization or MI maximization. In this paper, except the Simulation Experiment, we all select MI maximization as the objective. For reconstruction loss minimization, we use the mean absolute error (MAE) between our GCS outputs and entity representations of the knowledge-enhanced LMs. Regarding the MI maximization, we define the objective function by maximizing the MI as

$$\mathcal{L} = -\text{MI}(\text{GCS}_{\theta_1}(\mathbf{x}); \mathbf{h}). \tag{11}$$

We optimize MI equation 11 by maximizing the compression lemma lower bound (Banerjee, 2006) as in Belghazi et al. (2018). The inputs of GCS are $\boldsymbol{X}$, and let the output be denoted by $\boldsymbol{Z}$. We can regard $\boldsymbol{Z}$ and $\boldsymbol{H}$ as empirical samples of random variables $\mathbf{z}$ and $\mathbf{h}$. Thus, we have:

$$\text{MI}(\mathbf{z}; \mathbf{h}) \geq \sup_{T \in \mathcal{F}} \mathbb{E}_{\mathbb{P}_{\mathbf{zh}}}[T] - \log(\mathbb{E}_{\mathbb{P}_{\mathbf{z}} \otimes \mathbb{P}_{\mathbf{h}}}[e^T]). \tag{12}$$

Here, $\mathcal{F}$ can be any class of functions $T : \Omega \to \mathbb{R}$ satisfying certain integrability constraints (Belghazi et al., 2018). $\mathbb{P}_{\mathbf{zh}}$ represents the joint distribution of $\mathbf{z}$ and $\mathbf{h}$, and $\mathbb{P}_{\mathbf{z}} \otimes \mathbb{P}_{\mathbf{h}}$ represents the product of their marginal distributions. In practice, we let $\mathcal{F} = \{T_{\theta_2}\}$ be the set of functions parameterized by a neural network, and optimize it by stochastic gradient descent. Then, the objective function can be rephrased as

$$\max_{\theta_1, \theta_2} \left( \mathbb{E}_{\mathbb{P}_{\mathbf{z},\mathbf{h}}^{|\mathcal{V}|}}[T_{\theta_2}] - \log \left( \mathbb{E}_{\mathbb{P}_{\mathbf{z}}^{|\mathcal{V}|} \otimes \mathbb{P}_{\mathbf{h}}^{|\mathcal{V}|}}[e^{T_{\theta_2}}] \right) \right), \text{ where } \mathbf{z} = \text{GCS}_{\theta_1}(\mathbf{x}). \tag{13}$$

In equation 13, $\mathbb{P}_{\mathbf{z}}^{|\mathcal{V}|}$ represents the empirical distribution of $\mathbf{z}$, i.e., $\boldsymbol{Z}$. If the KG is very large, we can optimize the network by sampling a small subgraph of the KG. In practice, we simply add two MLPs layers to GCS for MI maximization. The added two MLP layers may not be bijective, where the dimension would be first reduced to $64$, then to $1$ for MI maximization. The nonlinear activation functions are all set as $\text{ELU}(\cdot)$ function, which is also bijective.

For interpretation, we use the attention coefficients on edges and self-loops to analyze the KI in terms of triples and entities. Different from Schlichtkrull et al. (2020) that specially designs a discrete function to mask edges that are not important, we simply introduce a temperature hyperparameter $t$ and set it as $t = 0.1$ to make the attention coefficient distribution hard[13]. Thus, knowledge can be well clustered into learned and unlearned.

## I  ADDITIONAL STATISTICS FOR INTEGRATION EXPERIMENT

**Table 7:** Drop statistics for the Integration Experiment.

| Datasets \ Statistics | Percentage of integrated entities | Percentage of integrated triples | # of aligned sentences/entity embeddings (integrated knowledge) |
|---|---|---|---|
| T-REx-rc | - | 28.86% | 561,687 out of 5,565,478 |
| Wikidata | 61.72% | - | 2,240,260 out of 3,275,534 |

## J  ADDITIONAL RESULTS FOR GCS VERIFICATION

We gradually drop out sentences aligned to knowledge triples whose attention coefficients are small for KI. For comparison, we randomly drop same number of sentences named the random strategy. We can see that dropping out KI datasets based on the interpretation results of GCS significantly outperforms the random strategy. It further supports that GCS can effectively interpret the KI situation for the K-Adapter.

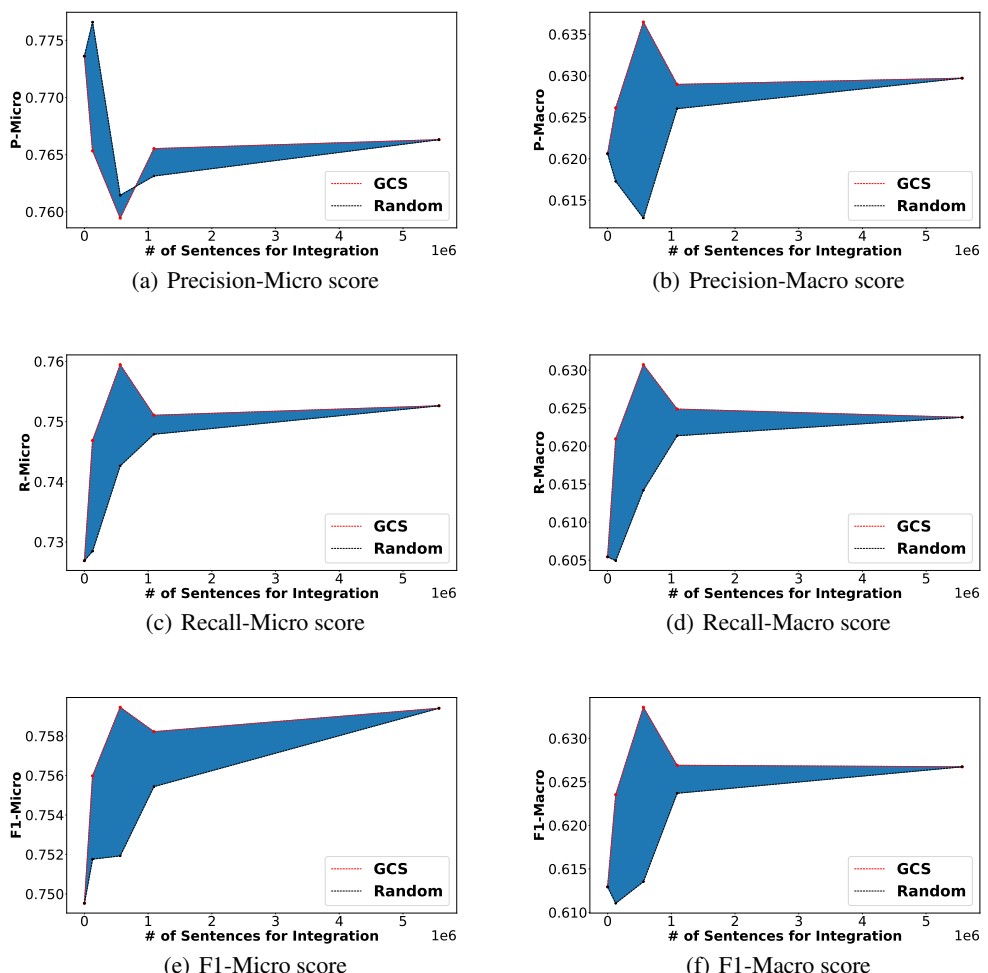

(a) Precision-Micro score

(b) Precision-Macro score

(c) Recall-Micro score

(d) Recall-Macro score

(e) F1-Micro score

(f) F1-Macro score

**Figure 4:** Detailed results of K-Adapter on the OpenEntity dataset, where we use different number of sentences to integrate knowledge into K-Adapter. For GCS, we integrate knowledge for K-Adapter using sentences aligned with triples whose attention coefficients are larger than $\{0.0, 0.01, 0.1, 0.9, 1.0\}$, where the corresponding number of sentences are $\{5565478, 1091152, 561687, 127728, 0\}$. Same number of sentences are chosen randomly for comparison as the random strategy.

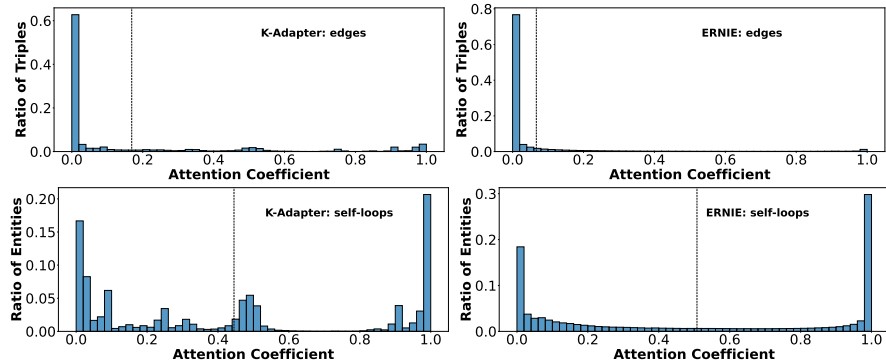

**Figure 5:** The attention coefficient distributions of edges and self-loops for K-Adapter and ERNIE. The histogram shows the empirical distributions (i.e., frequency), and the blue curves are the Gaussian kernel density estimate. The black dashed vertical lines indicate the average values.

## K    KNOWLEDGE INTEGRATION OVERVIEW

Figure 5 presents the empirical distributions of attention coefficients for K-Adapter and ERNIE. The above two subfigures show distributions on edges, interpreting the KI from triples. While the bottom two subfigures show distributions of self-loops, illustrating the KI from entities. We can find that most knowledge triples are not integrated well for both K-Adapter and ERNIE (i.e., $a_{i,j} < 0.1$), while K-Adapter performs slightly better. When it comes to entity-wise integration, in general, entity knowledge is also not integrated well. We find that both CR (i.e., $a_{i,i} > 0.9$) and CF (i.e., $a_{i,i} < 0.1$) happens for many entities, especially for ERNIE. K-Adapter outperforms ERNIE since some entities are integrated well (i.e., $0.4 < a_{i,i} < 0.6$).

## L    ADDITIONAL STATISTICS FOR TABLE 4

**Table 8:** The number of aligned sentences for relations.

| Statistics / Relation label | # of triples |
|---|---|
| Place of birth | 134,976 |
| Part of | 134,999 |
| Date of death | 135,190 |
| Date of birth | 135,169 |
| Located in the administrative territorial entity | 135,055 |
| Country | 135,147 |
| Total | 5,565,478 |

## M    ADDITIONAL EXPERIMENT: CASE STUDY

**Table 9:** The interpretation of KI, CR, and CF for K-Adapter in terms of relations and entities. We list 3 typical relations: CR happens to most connected entities; CF happens to most connected entities; and KI, CR, CF happens equally to connected entities. And we list 5 correspondingly aligned sentences. The Google Ngram of entities are reported to show the popularity of entities.

| Relation label | Ratio of triples connected to CR entities | Ratio of triples connected to CF entities | Ratio of triples connected to WL entities | # of aligned sentences |
|---|---|---|---|---|
| KI of Entities (Google Ngrams) | Examples | | | |
| Copyright license | 85.09% | 14.61% | 0.16% | 4,400 |
| CR, $(4.95, 2.47) \times 10^{-7}$ | This article incorporates **public domain material** from the United States Geological Survey document "Wasson Rock" (content from the **Geographic Names Information System**). | | | |
| Office held by head of government | 0.46% | 99.54% | 0.00% | 14,315 |
| CF, $(0.91, 1.33) \times 10^{-2}$ | Seven people served as governor of Colorado Territory over eight terms, appointed by the **President** of the **United States**. | | | |
| Opposite of | 26.30% | 41.87% | 28.62% | 11,577 |
| CR, $(1.18, 1.77) \times 10^{-4}$ | It is well covered with **deciduous** and **evergreen** forests. | | | |
| CF, $(3.67, 5.60) \times 10^{-2}$ | The prison's north wing is filled with **left**-wing rebels while the south wing is filled with **right**-wing government supporters and paramilitaries. | | | |
| WL, $(6.83, 2.46) \times 10^{-4}$ | Wildlife of Iran includes its **flora** and **fauna** and their natural habitats. | | | |

**Knowledge of different entities (sorted by their popularity) has different CR and CF ratios (case study).** Table 9 reports the KI for K-Adapter in terms of entities. We select three relations {"*Copyright license*", "*Office held by head of government*", "*Opposite of*"}. CR happens to most connected entities for relation "*Copyright license*". CF happens to most connected entities for relation "*Office held by head of government*". Some connected entities are learned well for relation "*Opposite of*". We list 5 aligned sentences for the three relations, and report the Google Ngrams (year 2019)[14] of entities to show their popularity. We find that CF often happens to entities with large Google Ngrams such as "*left*" and "*President*". And CR often happens to entities with small Google Ngrams. Well-learned entities are also not very popular.

---

[13]Note that the principle of hyperparameter selection is to maximize the MI, i.e., objective function. Users may select appropriate hyperparameters depending on the situation.

[14]https://books.google.com/ngrams

# N  ADDITIONAL EXPERIMENT: CORRELATION

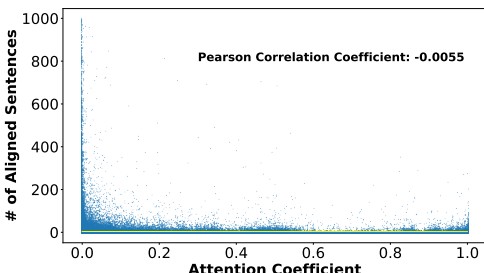

**Figure 6:** The correlation between the attention coefficient of the knowledge triple and its aligned sentence number. There is no correlation between them.

**Can we improve the KI quality by simply increasing the size of our dataset?** Above results analyze KI in LMs in different ways. However, there is a key question: can we simply improve the quality of KI by increasing the amount of our aligned training corpora? We try to answer this question in Figure 6 which plots correlation between the attention coefficients and the number of aligned sentences for knowledge triples in the dataset. We find that the *Pearson correlation* between the two is $-0.0055$. There is no positive correlation between the two variables. This implies there is no apparent positive relationship between the KI quality and the size of the KI dataset. It suggests that simply increasing the size of the aligned dataset alone may not improve KI, but, we might need more fundamental advances to push the state-of-the-art in KI forward.

