# OpenReview forum: "Understanding Knowledge Integration in Language Models with Graph Convolutions"
_ICLR.cc/2022/Conference — ICLR 2022 Submitted_

### Official Review · Reviewer_pRFd · 2021-10-31

**Correctness:** 4
**Technical Novelty And Significance:** 4
**Empirical Novelty And Significance:** 3
**Recommendation:** 8
**Confidence:** 3

**Main Review:**

The paper proposes to measure the amount of integrated knowledge with mutual information and shows the strong connection between the KI process and the Graph Fourier Transformation and then model it as graph convolution. The analysis process is interesting and coped with interesting insights.

A few comments on the paper:
1. Could you please clearly define 1-1, N-1, and M-N relations?
2. The authors uses attention weight on edges as measurement of relevance. It's okay to make the analysis using the attention weight in this paper, but it is arguable that attentions are not always explanations, i.e. "Attention is not Explanation" (Jain et al. 2019). Have you thought of performing you analysis using other measurement? Or maybe prove this is the exact measurement for your purpose?
3. When discussing "Can we improve the KI quality by simply increasing the size of our dataset?", how do you measure the "the number
of aligned sentences for knowledge triples in the dataset"? I might understand it wrong, but is the size of KB and text corpus in pretraining an important factor here. It maybe helpful to report the size of KB used in ERNIE and K-Adapter and modify it and see if it will change your conclusions.

Another work that may be interesting to consider:
[1] "Facts as Experts: Adaptable and Interpretable Neural Memory over Symbolic Knowledge" Pat Verga, Haitian Sun, Livio Baldini Soares, William W. Cohen

**Summary Of The Paper:**

The paper claims that the Knowledge Integration (KI) process can be simulated with Graph convolutions and designed a Graph Convolution Simulator (GCS) to analyze the performance of two popular KI models, K-Adapter and ERNIE. They observe that K-Adapter is good at capturing simple (1-1) relations while ERNIE is good at capturing complex relations. The experiments are well conducted and the conclusions are insightful.

**Summary Of The Review:**

This is an interesting analysis paper discussing the Knowledge integration process by proposing to measure their Mutual information (MI) and in turn model it as graph convolution. The claims are solid and thorough analysis are made. Good paper. Very enjoyable to read.

---

> ### Author Response · Authors · 2021-11-17
> **Reply to Reviewer pRFd**
>
> Thanks for the kind review. We will revise the paper as required and update the main paper ASAP. Let us first address your listed concerns one by one.
>
> 1. We’ll add clear definitions of 1-1, N-1, and M-N relations to the paper in red color. These relations are defined in TransE (Bordes et al., 2013). Simply speaking, 1-1 relations connect two leaf nodes (node degree=1). N-1 relations connect one leaf node and one center node (node degree>1). And N-M relations connect two centor nodes.
>
> 2. To support that our method (using graph attention for interpretation) could provide correct analysis results, we add one simulation experiment to compare it with distance-based methods and one linear classifier-based method. The results show that for KI analysis, our method provides reasonable results while baselines (distance-based methods and a classifier-based method) cannot provide meaningful results. Details can be found as follows.
>
>     **a. Simulation experiment: verifying GCS and baselines by simulations and comparing results of ERNE/K-Adapter with simulation results**. Specifically, we plan to add a simulation experiment where we set the input of GCS as noisy KG embedding (with different levels of noise) and the output as the groundtruth KG embedding (KGE), and study how probe models interpret the “KI process” (how much KG information is integrated from the noisy KGE to the ground-truth KGE). We select two kinds of **baselines** to interpret the “KI process”.
>
>     **Representation Analysis** (RA) methods: we calculate the Cosine/Euclidean similarities between two connected entities’ embeddings to see how it changes (Ideally, learning new KG info may make the similarity between entities larger).
>
>     **Probing** method: we design a linear classifier to do link prediction, and observe how its performance changes (Ideally, large positive gaps means that more KG information is integrated.)
>
>     The simulation results that will be updated in paper later show that **our method and RA methods can correctly show the “KI process”**: larger noise ratio would be interpreted with more integrated KG information. And the linear classifier fails to interpret when the noise ratio is large (>10%). Besides, we also set the input as BERT/RoBERTa entity embeddings and output as ERNIE/K-Adapter entity embeddings. We find that **only our GCS can provide meaningful results** (equivalent to noise ratio=5%): a small amount of KG information is integrated. For the baselines, results are not meaningful (RA: similarity becomes larger for K-Adapter and smaller for ERNIE, linear classifier: performance on ERNE/K-Adapter embedding is worse than BERT/RoBERTa).
>
> 3. The number of aligned sentences for knowledge triples in the dataset is based on the statistics of the T-REx-rc dataset. During the experiment, we don’t change that number. We only calculate the correlation between that number and the corresponding attention coefficient. Detailed statistics can be found in Table 6 (see Appendix E): T-REx-rc dataset (K-Adapter) has 1,282,504 triples and 5,565,478 aligned sentences.
>
> 4. For the references, we will read that paper and may include it in the related work.
>
> We hope that our response answers your questions and convinces you.

---

> ### Author Response · Authors · 2021-11-23
> **Reply for Reviewer pRFd**
>
> Dear reviewer,
>
> We would like to thank you for your time, and the opportunity to improve our paper. We have significantly improved the exposition in this paper and we have comprehensively performed all the analyses suggested by the reviewers and we have included the results of those experiments. Overall, we strongly believe that the writing modifications, results of the new experiments and our rebuttal should address many of the concerns raised in this review.  All the changes are summarized in our "Reply to all reviewers".
>
> Thanks,
> Authors of the submission

---

### Official Review · Reviewer_eLPV · 2021-11-02

**Correctness:** 3
**Technical Novelty And Significance:** 3
**Empirical Novelty And Significance:** 4
**Recommendation:** 6
**Confidence:** 3

**Main Review:**

Overall, the paper is well-written and presents a very useful method of analyzing how structured knowledge of entities may or may not be integrated into a language model through additional pretraining. The detailed analysis characterizing what kinds of knowledge are well-integrated, catastrophically remembered, or catastrophically forgotten based on relation complexity is helpful at demonstrating what kinds of probing the GCS method is capable of. Tying GCS to downstream task performance also helps to showcase the validity of the method, although this could be further strengthened by expanding on these results (see below).

One comment I had was regarding the presentation of the results in Table 1, which demonstrate that removing pretraining sentences based on knowledge identified as unlearned by GCS shows little drop in downstream task performance. While the results compare each of K-Adapter and ERNIE with their versions that drop certain pretraining examples, it would also be helpful to compare to BERT/RoBERTa results in the same table, as it is not immediately clear which differences between the two versions of each model are meaningful versus negligible without a baseline performance to compare to (though the random dropping results in the appendix somewhat help to clarify this). It would also be worth exploring downstream task performance in finer-grained detail to further support the method - for instance, how does an entity being catastrophically remembered/forgotten affect entity typing performance on examples that involve that entity? Correlating attention values found by GCS for specific entities to task performance on those entities would provide futher empirical validation of the method.

**Summary Of The Paper:**

This paper proposes Graph Convolution Simulator (GCS), a probing method for investigating what knowledge is being integrated into language models that are augmented with knowledge graph information through knowledge-enhanced pretraining. The method applies recent work for estimating the mutual information between continuous random variables using neural networks to capture the mutual information between entity representations before and after a language model has undergone knowledge integration. To make the method simpler and more interpretable, the mutual information estimation is relaxed to learning a combination of graph convolution and multilayer perceptron layers that map from the original entity representations to their knowledge-integrated versions, optimized by maximizing a compression lemma lower bound to mutual information from previous work. The graph convolution is parameterized as a graph attention layer, such that the attention weights can be used to quantify how much information is catastrophically forgotten or remembered for a single entity as well as transferred between entities that participate in a triple in the knowledge graph. This method is used to analyze two recent knowledge-enhanced language models, K-Adapter and ERNIE. The analysis shows that little factual knowledge is explicitly captured by these models, each model is better at capturing different kinds of relational knowledge (e.g., simple relational knowledge for K-Adapter, complex relational knowledge for ERNIE), and that K-Adapter struggles to remember temporal information and information about popular entities, among other findings. Finally, additional experiments show that increasing the size of the training corpora does not correlate with improved knowledge integration, motivating more careful development of future knowledge-enhanced pretraining methods.

**Summary Of The Review:**

While some changes to the presentation of results would further confirm the interpretability of the attention weights learned by GCS, the current set of results still seems to present a valid and useful method of probing for integrated knowledge in pretrained language models.

---

> ### Author Response · Authors · 2021-11-17
> **Reply to Reviewer eLPV**
>
> Thanks for the kind review. We report the performance of BERT and RoBERTa under the same setting (as ERNIE and K-Adapter). For the OpenEntity dataset (small dataset for finetuning), BERT performs 1% worse than ERNIE (2% worse than ERNIE-drop), and RoBERTa performs 0.8% worse than K-Adapter (0.8% worse than K-Adapter-drop). For the FIGER dataset (large dataset for finetuning), BERT performs 3% worse than ERNIE (2% worse than ERNIE-drop), and RoBERTa performs 1.5% worse than K-Adapter (1.4 worse than K-Adapter-drop). It could be more sufficient to support the claim that our GCS method correctly interprets the KI process.
>
> Besides, we analyze the test results of ERNIE and K-Adapter on the OpenEntity dataset. For K-Adapter, we find that in the test set, the performance on those entities that integrate much KG info is significantly higher than that on the forgotten entities. Specifically, we first align the entities in the OpenEntity dataset with entities in T-REx-rc (KI dataset for K-Adapter) and Wikidata (KI dataset for ERNIE) by their wiki Q identifier. Then, we run finetuning of K-Adapter and ERNIE on the OpenEntity training dataset, but test them with different test sets: one test set we remove all test data contain entities that are annotated as “well integrated” by the GCS (self-loop attention coefficient<0.1), another test set we remove test data that contain all entities that are annotated as “little information integrated” by the GCS (self-loop attention coefficient>0.1) (Note that there are also some entities that are not aligned, which means they are not in the T-REx-rc or Wikidata datasets). We find that the performance of ERNIE on the first test set is much worse than the original performance (>20%), and the performance on the second test set is slightly higher than the original one (1.6%). We are still running the experiment on K-Adapter, and will update the paper once we have the results.
>
> We hope that our response answers your questions and convinces you.

---

> > ### Comment · Reviewer_eLPV · 2021-11-28
> > **Reply to author response**
> >
> > Thank you for the response, and for the additional empirical results. While all suggested experiments seem to have been run, not all of the new results support the validity of the proposed method (e.g., the K-Adapter results in Table 2), so I will maintain my original score.

---

> > > ### Author Response · Authors · 2021-11-28
> > > **Reply to Reviewer eLPV's reply**
> > >
> > > Dear Reviewer eLPV,
> > >
> > > Thanks for your reply. We would like to explain why results of K-Adapter in Table 2 don't support the validity, and why the results are less important compared to ERNIE.
> > >
> > > 1. As mentioned in the paper (Section 4.2.3, paragraph 2): It may be because of the **differences in the finetuning objective and the KI objective**, and because the knowledge integrated in K-Adapter may change during finetuning. In Section 4.1 of the paper, we introduce that ERNIE integrates factual knowledge by aligning tokens in the input text with entities in KG. It is very similar to the downstream task (i.e., entity typing). However, K-Adapter integrates knowledge by deciding whether certain relations exist or not, and classifying relation labels given the aligned sentence. The KI objective is quite different from that of the downstream task. Note that our interpretation results is for KI. If finetuning task is very different, these KI interpretation results may not be well verified by the downstream task.
> > >
> > > 2. From Table 6, we can find that K-Adapter integrates knowledge by a small KG (subset of Wiki), while ERNIE integrates knowledge by a large KG. It means that **for entities in the OpenEntity's test set, many entities could not be aligned with the KI dataset of K-Adapter** (more than 20%). But for ERNIE, most entities are successfully aligned (more than 98%). Thus, compared to the results of K-Adapter, results of ERNIE are more persuasive.
> > >
> > > Overall, we thank you for the constructive suggestions about the experiment. KI interpretation is a difficult problem and our work proposes a new method. The added experiment can delicately verify our method from a different aspect.
> > >
> > > Best,
> > >
> > > The authors

---

> ### Author Response · Authors · 2021-11-23
> **Reply to Reviewer eLPV**
>
> Dear reviewer,
>
> We would like to thank you for your time, and the opportunity to improve our paper. We have significantly improved the exposition in this paper and we have comprehensively performed all the analyses suggested by the reviewers and we have included the results of those experiments. Overall, we strongly believe that the writing modifications, results of the new experiments and our rebuttal should address many of the concerns raised in this review.  All the changes are summarized in our "Reply to all reviewers".
>
> Thanks,
> Authors of the submission

---

### Official Review · Reviewer_4QcM · 2021-11-03

**Correctness:** 2
**Technical Novelty And Significance:** 3
**Empirical Novelty And Significance:** 2
**Recommendation:** 3
**Confidence:** 4

**Main Review:**

**Strengths**:
- The paper tries to tackle a very interesting question: how much knowledge is actually integrated by knowledge-enhancing PLMs, which knowledge and how can we best measure this.
- As far as I’m aware, the idea to distill a KI-simulating GCN using MI is original and it could be a neat idea if some concerns are addressed better.
- The paper is well-structured.

**Weaknesses**:
1. The paper is difficult to understand and reproduce. It is still not 100% clear to me what exactly is ultimately done in the approach, but also in the experiments, where it is not clear what exactly is being done in Table 2 (appendix F specifies the GCS architecture and interpretation principle, which have already been explained in the paper, rather than exactly explaining the experimental setup of Table 2).
2. Relation to related work: The authors cite Hou adn Sachan's work a few times but do not further indicate how it is related. In Hou and Sachan's paper, the goal is to measure how much of an *entire* linguistic structure's information (e.g. dependency parse), as a whole, is captured in PLM's representations, and they measure this using MI between an encoding of the linguistic graph and the PLM's representations. The goal in the submitted paper is similar (except applied to KG's, and measuring the gain in knowledge after KI), the GCS objective is similar to the Bird's Eye objective, except that it also trains the GCS model. Although the submitted paper extends to quantifying knowledge gain, there are parallels with Hou and Sachan, and the relation with that work should have been discussed better.
3. Lack of baselines: it is not clear to me (and not discussed in the paper) why some baselines have not been explored to measure the effects of KI, and relate the proposed method to. The change of factual knowledge could be measured by measuring local KG subgraph MI with PLM representations with and without KI using Hou and Sachan's method. More fine-grained information seems to have been addressed in Hou and Sachan too, in Section 2.4. Another baseline could be to see how easily every relevant KG triple can be reconstructed from the entity representations with (*h*) and without (*x*) KI. Without the authors discussing these possibilities and comparing to their approach, it is not clear why the presented approach should be chosen over more obvious possibilities.
4. I am not sure how much sense it makes to train a different network (which might have different properties) to maximize the MI with the outputs of a knowledge-enhanced PLM (thus in a way distilling it?), and using this derived model to analyze the properties of the original model. It is not clear to me why any of the conclusions made based on the "simulating" model should really apply to the original model. One issue for example is that the used GAT does not seem to incorporate relation types, whereas the base models do have access to that information. How closely can GCS then approximate the knowledge actually contained in these models? I think comparing the attention weights to the results of a baseline probe with and without KI would be necessary to convince a reader that the obtained attention weights can really be interpreted like that. Without this however, I find the claim of understanding KI, CR and CF insufficiently supported.
5. Unsatisfactory verification: GCS is only verified on two entity typing datasets. The verification is done by using GCS to identify which knowledge is "learned", removing the knowledge that is **not** flagged as learned and then retraining the base models (ERNIE and K-Adapter). The authors note that the performance on the tasks stays more or less the same, which is taken for proof that GCS really identifies the knowledge that is actually learned. The authors also compare this to randomly dropping knowledge. Even though the comparison to random is crucial to put the numbers in context and verify the claim, the authors put these numbers in appendix. While there indeed appears to be a higher performance penalty when dropping randomly at a certain drop rate (although the difference is only 0.006 (0.6%) micro F1 and 0.03 macro F1 points at its largest), only results for K-Adapter on OpenEntity are given (in appendix). This is only one of the four settings reported in Table 1. More complete numbers are necessary to properly judge the claim.
In addition, I believe that a better verification method needs to be developed where the absence of knowledge is more penalizing (in Figure 6, training without any KI doesn't appear to be much of a problem).
Another issue is that the baseline performance of K-Adapter in Table 1 is lower than that reported in the K-Adapter original paper, and even worse than the RoBERTa baseline used in that paper.

**Questions** and **Remarks**:

6. How are you taking into account the different relation types when doing GAT in GCS?

7. Remark: I think it should be clarified what is simple and what is complex knowledge in the beginning.

8. Remark: It should be clarified what is meant by "entity text corresponding to the node v_i". I assume this is the entity label. However, as it is written, it could also be an entity description, or any kind of text, that could also describe entity properties.

9. Probably AutoPrompt (Shin et al. 2020) and OptiPrompt (Zhong et al. 2021) should be cited as well.

10. The quality of the writing should be improved. Examples:
- “which aim of” → “which aim to”
- “The transformation is a simulation of the KI process, i.e., MI change, and it promise the accuracy.” ← this sentence does not make sense to me
- “Below two” → “bottom two”
- Some specs in Fig. 6 description are basically unreadable because of commas everywhere.


**Summary Of The Paper:**

In this paper, the authors investigate a novel approach to improve our understanding of the knowledge integration (KI) process of knowledge-enhanced pretrained language models (PLM). While several works (Petroni et al. 2019 etc) have investigated knowledge contained in PLM, in this work, the focus is to understand how much and what kinds of knowledge are actually captured by methods that take PLM's and perform addition pretraining to incorporate factual knowledge from knowledge graphs (KG) into the language models.
The authors interpret the gain in mutual information (MI) between the PLM's text representations and knowledge graphs as a measurement for KI. The authors note that MI is difficult to estimate and note that instead the KI process can be simulated using a transformation that relies on forward and reverse graph Fourier transforms and a NN in that domain. However, since this can be expensive, the authors propose to instead use graph convolutions.
Thus, the proposed method attempts to "simulate" the knowledge integration process as graph convolution (GC) over the knowledge graph and is called Graph Convolution Simulation (GCS). The authors choose a model based on attention-based GC (graph attention network, or GAT) in order to improve interpretability. The actual model finally used by the authors consists of two MLP layers and one GAT layer in between. The GCS model is trained by maximizing the mutual information (MI) of the entity representations of GCS and the entity representaitons obtained from the PLM.
The authors then use the attention weights of the GAT to determine what information from the KG is used by the simulation in order to maximize its MI with the actual representations built by the knowledge-enhanced PLM. According to the authors, this allows to investigate catastrophic forgetting and remembering and which relations and types of relations are learned well during KI.
In experiments, GCS is first verified on entity typing. Subsequently, GCS is used to investigate what information is learned by ERNIE and K-Adapter.

**Summary Of The Review:**

=== **after rebuttal** ===

While the theoretical discussion appears sound, I do not think that contribution in itself warrants acceptance at a venue like ICLR. I have an issue with the expressivity of the proposed implementation of GCS since it does not encode relation type information, which is essential in the definition of a triple.

However, my main concerns are with the experimental verification part, where the claimed ability of the proposed method to identify which knowledge is learned is not acceptably verified and the experiments may suffer from other issues.

While I appreciate the efforts of the authors, and think the paper now looks better, I think it does not deserve publication at ICLR in its current form and I will not be raising my score. My concerns with some critical issues of this work were not resolved by the author's reply and the updated paper.

=== **before rebuttal** ===


While the paper focuses on an interesting question, due to the above issues, in my current understanding, I don't think it deserves publication in its current form at a venue like ICLR. While it's different from previous work that I'm aware of and might be a neat idea, I found the claim of using attention weights of a GAT-based network simulating the KI process to inspect what is being learned not sufficiently justified. In my opinion, the proposed simulating probe has not been properly verified, and it is not clear why it has not been compared to more basic methods and straightforward adaptation of previous work.

---

> ### Author Response · Authors · 2021-11-17
> **Reply to Reviewer 4QcM (part 1/3)**
>
> Thanks for the constructive review. Currently we are trying to add more experiments to verify our GCS model. We will revise the paper as required and update the main paper ASAP. Let us first address your concerns one by one.
>
> 1. Clarity and reproducibility. We will reorganize the paper to emphasize our theoretical contribution and make it more readable. As for the approach, our GCS model simulates the change of entity representations (from BERT/RoBERTa to ERNIE/K-Adapter), and the attention coefficients on edges of KGs are used for interpretation. The purpose of our approach is to interpret the KI process (e.g., how much KG info is integrated, which set of triples are learned) for different KI methods. The procedure is that we first simulate and interpret the KI process with results (attention weights), then we analyze the results for conclusions. In Table 2, the interpretation step has been done (in section 4.4), and conclusions are sorted and shown in Table 2. Appendix F illustrates how we simulate the KI process with GCS, which explains how the results in Table 2 come from. As for the reproducibility, we will publish the code as well as the results (interpretation results, the reproducibility of ERNIE and K-Adapter, etc.). Besides, we argue that it is simple to reproduce the analysis with GCS. Only two MLP layers with one GCN layer (with graph attention) composed of  the GCS model, the input is entity representations of BERT/RoBERT (& KGs) and the output are entity representations of ERNIE/K-Adapter. After training the GCS model with loss as -MI(output, entity representations of ERNIE/K-Adapter) or the reconstruction loss, we can get attention coefficients on edges of KGs for interpretation. The hyperparameters of GCS are also provided in the Appendix F.
>
> 2. We will add it in the main paper and upload it ASAP. Regarding the related work (Hou & Sachan), the purpose of their paper is to probe linguistic knowledge in one LM, while our purpose is to probe factual knowledge change during the KI process (from LM to knowledge-enhanced LM). The only reason why we cite their paper is that they claim that linguistic knowledge can be measured by the MI. And we measure the KI process by the MI change. To interpret how KG (i.e., knowledge triples) contribute to that change, we use GCS to simulate that change and use its graph attention on edges (i.e., triples) for interpretation. The objective of GCS is to let GCS’s output be close to entity representations of ERNIE/K-Adapter (in an information-theoretic view). And thus based on the different purposes and methods, our work is not very relevant to the related work (Hou & Sachan). If we simply extend and implement their work for KI interpretation, i.e., large KGs instead of small linguistic graphs, the quality of graph embedding will be quite bad (Even if KG embedding algorithms such as TransE cannot achieve good performance as the bijective function.), and thus the MI estimation could be very noisy for interpretation.

---

> ### Author Response · Authors · 2021-11-17
> **Reply to Reviewer 4QcM (part 2/3)**
>
> 3. As for the baselines, implementing them in the KI scenario is challenging, since probing KGs is different from probing small linguistic graphs. The large sparse KG makes it troublesome to design probe models. For example, using MI method in the related work (Hou & Sachan). we think that it is nearly impossible to implement it in practice in the KI scenario (as also mentioned in point 2.). First, they use deepwalk to embed graph structures. It performs well for small linguistic graphs (less than 50 nodes and 100 edges). But for the KG, the scale is much larger than linguistic graphs, millions or even billions of entities and relations are integrated. Embedding KGs would cause much structure information loss even using SOTA KG embedding methods. Thus, their theoretical assumption cannot hold, and the results could not correctly interpret the KI process. Second, efficiency. With our GCS model, we can simulate the KI integration only one time, and results for all triples and entities are obtained. However, if we simply implement Hou & Sachan’s method for large KGs (e.g., Wikidata for ERNIE) to probe local structures, each time we drop a set of triples/entities or a local graph, we have to re-estimate the new MI values before the interpretation. Besides, as also mentioned in their paper, MI estimation is quite noisy. Even if they propose two control bounds, they still need to repeat MI estimation for many times to reduce variance. And that makes it impossible to probe small set of triples/entities delicately. Even if their method is not feasible to be a baseline, we still try our best to evaluate our method empirically. Detailed settings and results will be updated in the paper ASAP.
>
>     **a. Simulation experiment: verifying GCS and baselines by simulations and comparing results of ERNE/K-Adapter with simulation results**. Specifically, we plan to add a simulation experiment where we set the input of GCS as noisy KG embedding (with different levels of noise) and the output as the groundtruth KG embedding (KGE), and study how probe models interpret the “KI process” (how much KG information is integrated from the noisy KGE to the ground-truth KGE). We select two kinds of **baselines** to interpret the “KI process”.
>
>     **Representation Analysis** (RA) methods: we calculate the Cosine/Euclidean similarities between two connected entities’ embeddings to see how it changes (Ideally, learning new KG info may make the similarity between entities larger).
>
>     **Probing** method: we design a linear classifier to do link prediction, and observe how its performance changes (Ideally, large positive gaps means that more KG information is integrated.)
>
>     The simulation results that will be updated in paper later show that **our method and RA methods can correctly show the “KI process”**: larger noise ratio would be interpreted with more integrated KG information. And the linear classifier fails to interpret when the noise ratio is large (>10%). Besides, we also set the input as BERT/RoBERTa entity embeddings and output as ERNIE/K-Adapter entity embeddings. We find that **only our GCS can provide meaningful results** (equivalent to noise ratio=5%): a small amount of KG information is integrated. For the baselines, results are not meaningful (RA: similarity becomes larger for K-Adapter and smaller for ERNIE, linear classifier: performance on ERNE/K-Adapter embedding is worse than BERT/RoBERTa).
>
> 4. The intuition behind the method is based on the formulation of the KI process (also mentioned in point 2). It is similar to general probing papers, which design a probe model, e.g., linear classifier to interpret LMs. We first define the random variable of KG and LM, and use the MI change to define the KI process. And from the theoretical analysis in section 2, we know that the KI process can be approximated by a transformation, and the MI change only happens in the linear function in the graph Fourier space. Then,we get the conclusion that the KI process (complex and difficult to analyze) is “equivalent” to the graph convolution operation (simple and interpretable). This theoretical conclusion motivates us to design the GCS model to simulate and interpret the KI process. To prove that conclusions made based on GCS can be applied to the KI process (from original LM to knowledge-enhanced LMs), we provide analysis with proofs theoretically. Besides, we also add one simulation experiment (described in point 3.a) to further support it empirically. For relation type (see point 6.).

---

> ### Author Response · Authors · 2021-11-17
> **Reply to Reviewer 4QcM (part 3/3)**
>
> 5. For the verification of GCS, we add one simulation experiment (described in point 3.a). Besides, we report the reproduced performance of BERT/RoBERTa under the same setting to complement that experiment. Third, we add one analysis experiment on the downstream task as Reviewer eLPV required. Let us elaborate those experiments and address your concern here. First, after we add the simulation experiment and the analysis experiment on the downstream task, the verification of GCS is more comprehensive. As for the comparison to the random strategy, we only report results for two reasons:
>
>     **a.** The experiment is too costly. For each random drop strategy, we have to repretrain ERNIE/K-Adapter (KI process), and then finetune them on different downstream tasks. It is very costly, and may take months to get the results. And verifying GCS by performance on downstream tasks is not a perfect way: new knowledge in the training data (finetuning) may be integrated, and thus the performance could be not very consistent with our analysis results (about KI process).
>
>     **b.** Fair experiment settings are hard to promise. We know that finetuning is troublesome and overfitting can easily happen. To verify GCS by downstream tasks, we should set LMs’ hyperparameters be the same and make sure overfitting doesn’t happen to all LMs. When the number of LMsis large, we may need to rerun obtained results multiple times to make sure they have the same setting.
>
>     That’s why we only run experiments for K-Adapter on OpenEntity. However, we still try our best to provide more results to verify GCS.
>
>     **c.** We will research the results of the downstream task (entity typing on OpenEntity dataset). For K-Adapter, we find that in the test set, the performance on those entities that integrate much KG info is significantly higher than that on the forgotten entities. Specifically, we first align the entities in the OpenEntity dataset with entities in T-REx-rc (KI dataset for K-Adapter) and Wikidata (KI dataset for ERNIE) by their wiki Q identifier. Then, we run finetuning of K-Adapter and ERNIE on the OpenEntity training dataset, but test them with different test sets: one test set we remove all test data contain entities that are annotated as “well integrated” by the GCS (self-loop attention coefficient<0.1), another test set we remove test data that contain all entities that are annotated as “little information integrated” by the GCS (self-loop attention coefficient>0.1) (Note that there are also some entities that are not aligned, which means they are not in the T-REx-rc or Wikidata datasets). We find that the performance of ERNIE on the first test set is much worse than the original performance (>20%), and the performance on the second test set is slightly higher than the original one (1.6%). We are still running the experiment on K-Adapter, and will update the paper once we have the results.
>
>     As for the issue that our reproduced K-Adapter is even worse than RoBERTa, our purpose is to get relative performance instead of finding best hyperparameters for higher performance. Thus, we need to set all the settings as the same (see point 5.b) to make the comparison fair.  We also reproduce their KI process to ensure that. We report the performance of BERT and RoBERTa under the same setting (as ERNIE and K-Adapter). For the OpenEntity dataset (small dataset for finetuning), BERT performs 1% worse than ERNIE (2% worse than ERNIE-drop), and RoBERTa performs 0.8% worse than K-Adapter (0.8% worse than K-Adapter-drop). For the FIGER dataset (large dataset for finetuning), BERT performs 2% worse than ERNIE (2% worse than ERNIE-drop), and RoBERTa performs 1.5% worse than K-Adapter (1.4 worse than K-Adapter-drop).
>
> 6. In this paper, we consider a general method without too much side information. The complex KG with different relation types, entity descriptions, multi-hop relations, and time stamps are left for future work.
>
> 7. We will reorganize the paper to revise that.
>
> 8. Indeed it is the entity label. We will clarify that.
>
> 9. We will add those two papers as citations.
>
> 10. We will reorganize and polish the paper.
>
> We hope that our response answers your questions and convinces you.

---

> ### Author Response · Authors · 2021-11-23
> **Reply for Reviewer 4QcM**
>
> Dear reviewer,
>
> We would like to thank you for your time, and the opportunity to improve our paper. We have significantly improved the exposition in this paper and we have comprehensively performed all the analyses suggested by the reviewers and we have included the results of those experiments. Overall, we strongly believe that the writing modifications, results of the new experiments and our rebuttal should address many of the concerns raised in this review.  All the changes are summarized in our "Reply to all reviewers".
>
> Thanks,
> Authors of the submission

---

> ### Author Response · Authors · 2021-11-26
> **Reply to Reviewer 4QcM (any new concerns?)**
>
> Dear reviewer 4QcM,
>
> Thank you again for your valuable comments! We want to know if your concerns have been properly addressed. If you have any other (unresolved or new) concerns after reviewing our feedback and paper, please let us know. We are very happy to answer them.
>
> Best,
>
> The authors

---

> > ### Comment · Reviewer_4QcM · 2021-11-26
> > **Reply to author reply**
> >
> > Hi,
> > Thank you for the elaborate reply and for taking effort to address our comments!
> > Here’s some abbreviations for the response
> > KE LM = knowledge-enhanced language model
> >
> > I. Regarding the simulation experiment, it should be explained better and more formally for easier and clearer understanding. It should also be better explained why you’re doing it. Figure 3 just says “interpretation results”. What metrics are actually reported there? Where are the details about the linear probe and how it has been trained (Appendix F just briefly goes over it without any more concrete details)? It is not clear how strong of a baseline it really is, and whether differences in AUC (?) are comparable to the other numbers.
> > Another question is why do you expect that the entity representations of neighbouring entities must be similar in order for the fact to be learned, especially for KE LMs? ERNIE takes TransE embeddings, which are not encouraged to be similar.
> >
> > I don’t think this section is ready for publication. Even though this analysis may be valid or at least going in the right direction, much more details are needed to be able to judge it properly.
> >
> > As a sidenote, I think Fig. 3 will be difficult to read on paper, and might be problematic for colorblind people (please check guidelines for next iteration).
> >
> > Re: 4: I still don’t understand how valid the GAT simulation without relation types is. Unlike entity descriptions and time stamps, relation types encode crucial information about facts (being “born in”, Zurich is very different from being “died in”, Zurich). Without taking into account relation types, I don’t understand how it can be argued that the simulating GAT can interpret the KI correctly in terms of which facts are learned (it might be increasing MI by using different facts) and another question is how close can it actually get. For example, ERNIE integrates TransE embeddings, and those have been built with the use of relation types.
> >
> > Re: 5: I think the results on entity typing verification should be complete, but I don’t think it’s necessary to re-run the KE pre-training for all those thresholds, just for the one you are comparing in Table 1. Comparison to the random strategy is crucial to put into perspective the impact of dropping based on GCS. I think without it, the numbers currently present in Table 1 don’t say much: if randomly dropping 90% of the sentences, pre-training on the remaining ones, and then fine-tuning obtains performance very similar to dropping 90% of the sentences using GCS, it might mean that GCS doesn’t properly identify learned facts (which would be a negative result and motivate to dig deeper), or that the choice of which facts are dropped for KE pre-training isn’t really important to the downstream task (which would mean the evaluation is meaningless). If the numbers can’t be provided now, I would suggest resubmitting the paper later.
> >
> > I’m still not sure why a decrease in F1 of the dropped version is considered a good thing (assuming green is good)? Doesn’t it mean that some of the facts you removed were actually learned by the KE LM and were useful for the entity typing task but GCS didn't think they were learned? In fact, I’m also not sure what to make of an increased F1 for dropped versions. The dropped sentences resulted in added knowledge that was actually counter-productive? And doesn’t a positive change also mean that some facts were dropped that were actually learned by the KE LM (even though they were counter-productive)? So shouldn’t *any* deviation from non-dropped versions be considered bad?
> >
> > Re: 5. reproduction: I’m not sure just looking at relative performance is convincing when the baseline (and the KI-models) is so much lower than reported by others. ERNIE’s performance is 75.56 micro F1 on OpenEntity (compared to 73.19 here), and 73.39 on FIGER (compared to 71.13 here). K-Adapter had 77.61 on OpenEntity (compared to 75.94 here) and 80.54 on FIGER (compared to 76.69). I’m not sure such large differences between your experiments and theirs should be dismissed and at the very least, I think these originally reported numbers should be reported in the table. Also, I don’t see the problem (regarding fairness or computational effort) with starting from the settings used by K-Adapter in order to train a dropped K-Adapter and doing the same for RoBERTa, BERT and ERNIE. K-Adapter code is available at https://github.com/microsoft/k-adapter, and the hyperparameters seem to be provided in the scripts.
> >
> >
> >
> > I realize some of my questions are quite late considering the deadline, but I would appreciate to hear your thoughts on them.

---

> > > ### Author Response · Authors · 2021-11-27
> > > **Reply to Reviewer 4QcM's reply (Part 1/2)**
> > >
> > > Dear Reviewer 4QcM,
> > >
> > > Thanks for your intime response. We’ll try to answer your questions point by point:
> > > We begin by describing our theoretical contribution. In this paper, we theoretically formulate the KI process, and propose that it can be simulated and interpreted by graph attention in graph convolutions (Please see Definition 1, Theorem 2, Proposition 3, and Proposition 4 to get the theoretical intuition behind the our implemented GCS model.). This is one of our core contributions. Understanding the theoretical part correctly can definitely help you get the point of this paper and may address some of your concerns directly. Then, we design GCS based on our theoretical conclusions, and verify this model by designing a set of experiments. Besides, we use GCS to analyze K-Adapter and ERNIE and get some interesting findings.
> > >
> > > Let’s then answer your questions point by point.
> > >
> > > 1. We are sorry for the rushed update. We will keep polishing the writing in this subsubsection to make it clearer. In the simulation experiment, we compare different methods on interpreting KG information in LMs. Considering that there is no existing work about it, we have to design the simulation experiment and extend existing representative model explanation methods as baselines. The key point here is to prove that simply extending existing methods can’t interpret KI well, and thus we have to design a new method.
> > >
> > >     * 1.1 The metrics depend on the method (Section 4.2.1, paragraph 2). For RA (representation analysis methods), we simply calculate the Cosine&Euclidean similarity change between two connected entities (Appendix F, paragraph 3 shows the details).
> > >
> > >     * 1.2 For the linear probe, it is a linear classifier, and it’s trained by the link prediction task with negative sample number = 5. All the details as well as the training and test data split details are provided in Appendix F. Other hyperparameters are exactly the same as GCS (e.g., lr, optimizer). We believe that this linear probe is easy to implement and it’s widely used in probe models, and we have already tested that different hyperparameters (e.g., lr, optimizer) won’t influence the results much. We will add more details about it in the appendix.
> > >
> > >     * 1.3 For the baselines, note that there is little discussion about the interpretation of KI. This is because there is no existing work that can tackle this well. The reason is that we do not obtain good probes by simple extensions to existing probes. The link prediction task always faces the class imbalance problem especially for large and sparse KG. We select two representative interpretation methods as baselines (representation analysis methods and linear probe) to show that existing probe methods cannot provide any reasonable interpretation.
> > >
> > >     * The reason why we report AUC is that F1-score and accuracy totally become meaningless (F1 score is always close to 0, and acc is always close to 1) for the imbalance classification task (link prediction), but AUC score is distinguishable since it’s often used to measure the performance of link prediction. Thus, the difference of AUC is comparable to other numbers.
> > >
> > >     * 1.4 For the RA methods, we calculate the similarity change between entities as the baselines. The intuition behind is common in the link prediction task: similar nodes (entities) tend to be connected together. It can also be regarded as that we use k-Nearest Neighbors (kNN) to do the link prediction task (using Cosine and Euclidean similarity as metrics).
> > >
> > > 2. We will keep polishing this subsubsection to make it more readable. We argue that for the complex KI interpretation task, there is no existing work that can get proper interpretation results. It’s not possible to find more justified probe baselines in this condition (see point 1.3).
> > >
> > > 3. For Figure 3, we will adjust the plot based on the guidelines for the next iteration.
> > >
> > > 4. For the side information on relations, such as relation direction, relation type, relation label, indeed they are important. Based on the proposition 3 & 4, the side information can be easily supported by different graph filters. Simply speaking, RGCN with graph attention can be selected to support different types of relations. We would like to leave that exploration for future works.

---

> > > > ### Comment · Reviewer_4QcM · 2021-11-29
> > > > **Reply to reply**
> > > >
> > > > Thank you for the elaborate reply!
> > > > - 1. The information provided in your reply was already provided in the paper, i was hoping to see more and formal details so I'm afraid it doesn't clarify much.
> > > > I'm still not convinced the used metrics are even comparable on the same scale. From what I understood, you are comparing raw differences in cosine similarities (RA), AUC (probe) and attention (?) scores.
> > > >
> > > > - 4. I have no doubt relation types can easily be supported, I am concerned that here you chose to simply ignore this information, while it is essential for the definition of a triple.
> > > >
> > > > - 5. Without comparison to **anything** with access to similar amounts of information, I do not understand under what conditions the used test that tests whether GCS can identify learned facts is considered as failed. Where do you draw the line and say "our method was actually *not* able to identify the learned facts"? So I am not convinced about the presented verification in this section. Also, how much was the variance over seeds?
> > > >
> > > > - 6. First, I don't understand why you even consider "carefully selecting seeds" for anything (I guess you didn't for any results in the paper, but I don't know why you report numbers with seed tuning in your reply). I would have liked to see the best possible performance (averaged over carelessly chosen random seeds) of all methods on both datasets because the hyperparameters that work best for one method may not be the optimal for another method. I believe the numbers for the different methods should be obtained independently from each other and then compared, while staying on the same hyperparameter search budget (e.g. 100 different hyperparameter settings tested for each method). What were the best results for RoBERTa, K-Adapter and K-Adapter (dropped) under these conditions? This is not about finding the best hyperparameters to squeeze out that extra percent and beat the state-of-the-art one more time, but to ensure fair search conditions, which is essential for fair comparison between alternatives.
> > > > If you used their code and hyperparameters, I still don't understand why your numbers are so much lower.

---

> > > > > ### Author Response · Authors · 2021-11-29
> > > > > **Reply to the last reply**
> > > > >
> > > > > Dear Reviewer 4QcM,
> > > > >
> > > > > Thanks for your intime response. Let's try to answer your new questions one by one.
> > > > >
> > > > > 1. We're sorry for the misunderstanding. Here, we are not comparing the absolute values of different metrics. Our foucs is the tendency of curves (Due to the space issue, we put them in one figure. We'll revise that and make it clearer). Ideally, for all methods (with different metrics), with the increase of noise ratio, the curve should increase as well. However, we found that for linear probe, the tendency doesn't behave as expected. This means that the linear probe can't correctly interpret the sythetic KI process.
> > > > >
> > > > >     For the results of each method, we know that these sythetic KI processes range from integrating nothing from KG to all info from KG. In practice, the results for K-Adapter and ERNIE should belong to the scope (integrating some knowledge from KG). However, for the practical results of all baselines, they are not in the scope. Thus why we say only GCS provides reasonable results.
> > > > >
> > > > > 4. We understand the concern about the definition. Indeed, the relation type is very important for knowledge triples. But note that for KI, instead of entity representations, the relation information (i.e., relation type, relation direction) remains the same for vanilla LMs and KE LMs (i.e., during KI). Thus, we decide to focus on entity representations first and leave the extension for future work.
> > > > >
> > > > > 5. We understand the concern here. The question can be answered in Table 2. We compare the performance of downstream tasks for K-Adapter and ERNIE on the learned knowledge and unlearned knowledge, and find that they performs obviously better on the test data related to learned knowledge. You may regard the performance of unlearned knowledge as "random control group". And it may prove that GCS doesn't fail to identiy learned knowledge.
> > > > >
> > > > > 6. We understand the concern here. For the performance on downstream task, we have to notice that the test set is quite small (only 2 thousand samples). And thus different from other popular applications, the performance is very unstable. That's the reason why we mentioned that random seend can even influence the performance. As for the fairness of comparison, we agree with that the method proposed by the reviewer is also reasonable. But please note that our work is not to propose a better KELMs. For their drop versions, we use same architecture, same KI settings, and same code (published by their authors) as K-Adapter and ERNIE. The only change here is the size of input KI data. Comparing to the work whose contribution is proposing a new KELMs, using the same settings in our work is obviously a better choice.
> > > > >
> > > > >     As for the worse performance issues, we do not question the results in K-Adapter and ERNIE, since their models indeed have some improvement compared to vanilla LMs. We don't know whether they miss mentioning some steps on their github repos. But running their code as they instructed cannot get the same score as they mentioned in papers. And it is not our responsibility to 100% reproduce their results, considering that we run all experiment by their provided code, didn't change anything except the fp16 to fp 32.
> > > > >
> > > > > Overall, we thank you for your time and detailed comments. They are helpful for us. And hope our answers are also helpful for you to address aforementioned concerns.
> > > > >
> > > > > Thanks,
> > > > >
> > > > > the authors

---

> > > ### Author Response · Authors · 2021-11-27
> > > **Reply to Reviewer 4QcM's reply (Part 2/2)**
> > >
> > > 5. We provide an argument to show why the random strategy is not crucial. Consider ERNIE as an example, it successfully integrates 60% entities as GCS interprets. Assume that there are 100 entities, 60 entities are correctly integrated. And our GCS model correctly interprets 50 entities out of 60 (Note that GCS can't give 100% correct interpretation.). Now, if we randomly sample 60 entities, the expected positive number is 36. Note that there is also variance in the random strategy: P(#>=40) ~= 0.1. As the reviewer agrees, we are handling the KI (also can be regarded as secondary pretraining). It is impossible to repeat the pretraining for multiple times to reduce the variance. Even if we can do that, the expected positive number of the random strategy is not significantly smaller than the groundtruth (36 vs. 60). Thus, we do not think that comparing with random strategies is crucial for verification.
> > >
> > >   * Besides, our GCS model is designed for KI analysis, instead of finetuning analysis. As we know that finetuning step would change the LM distribution. This introduces extra variance to the performance of random strategy. Moreover, we use downstream task performance for verification since this is a new problem and there is no other better way (e.g., comparing with existing works).
> > >
> > >
> > >   * As for the decrease in F1, we note that GCS model can't provide 100% correct results. So, GCS misinterprets some knowledge integration cases (see the specified decreased percentage in Table 1). And for the positive change, we suppose it is because removing knowledge that can’t be learned during KI may alleviate the catastrophic forgetting problem. The LM thus may keep more linguistic knowledge learned in the pretraining step. Besides, there is some randomness during finetuning: e.g., if we randomly shuffle the training data, results could also slightly fluctuate.
> > >
> > > 6. Thanks for sharing your concerns about reproducibility. We will publish code and intermediate results after the review process has finished (we still need some time to sort code and results).
> > >
> > >   * Below, we explain why we can’t use their reported number directly and why the performance of K-Adapter and ERNIE is lower than they report.
> > >
> > >   * For KI, we reproduce their KI process exactly the same as introduced in their github repos. The only difference is that we use fp32 instead of fp16. We will also publish our reproduced K-Adapter and ERNIE models. Anyone interested can reproduce the KI process easily.
> > >
> > >   * For the reported performance of K-Adapter and ERNIE. If we report numbers given by their paper, there are three issues. First, the finetuning step could be time-consuming, and the authors of K-Adapter and ERNIE didn’t (and couldn't) thoroughly search the best hyperparameters. We tried on the OpenEntity dataset (which is small) and found that we can finetune RoBERTa to 78/79 F1-Micro if we carefully select random seeds and hyperparameters (1-2% higher than the reported score in their paper.). Second, related to the first point, we can always search for better hyperparameters/random seeds with a large scope using this unclean trick, and easily beat the original versions, no matter how much knowledge is integrated. Thus, after careful consideration, we decide to use their provided hyperparameters for experiment, and set all the settings for all LMs as the same.
> > >
> > >   * For finetuning, we make sure every model has the same setting. As our focus is not to find best hyperparameters for all models, our focus is to compare KI in a fair setting and make sure there are as few influence factors as possible. We suppose that the performance is worse because of three reasons.
> > >
> > >     * a) Some tricks about fp16 in pretraining: check --loss_scale in ERNIE (stable pretraining), and --fp16_opt_level in K-Adapter (mixed precision).
> > >
> > >     * b) Some tricks in finetuning. For example, for the OpenEntity dataset in K-Adapter repo, they said “it takes about 2 hours to get the best result running on single 16G P100”. However, we try to use P100 to reproduce it, but find that 2 hours is totally not enough since they set “per_gpu_train_batch_size=4”. Thus, the author may use some early stop strategies to select the best performance to report in their paper. But we all use the final output as the results (by running their provided code).
> > >
> > >     * c1) The version of K-Adapter used in our experiment is K-Adapter (fac). We don’t consider the implementation of the linguistic adapter.
> > >
> > >     * c2) For ERNIE, the KI dataset (Wikipedia version) used in the paper is not clearly introduced. Thus, we simply run their code as instructed, and the code fetches the latest version of Wikipedia and uses it for KI.
> > >
> > >
> > > We hope this reply helps you appreciate our work. While our presentation has some loose ends, we are improving it. We believe it is going in a new direction where it is hard to easily present experimental conclusions. Hope you still find our theoretical and empirical contributions valuable.

---

### Official Review · Reviewer_WDUZ · 2021-11-04

**Correctness:** 3
**Technical Novelty And Significance:** 2
**Empirical Novelty And Significance:** 3
**Recommendation:** 5
**Confidence:** 4

**Main Review:**

Strengths:
- A clear and intuitive method to understand the knowledge integration process of LMs by estimating the MI with entity representations via applying GFT and attention.
- The findings with GCS about the factors such as the popularity of entities and the size of KI dataset are beneficial for future research in KI.

Weakness:
- The study is quite limited to two particular KI methods and it's not convincing that the GCS method can be applied to analyze general KI methods and lead to correct conclusions.
- The method is limited to binary relational facts (subject, predicate, object) while real-world scenarios of fact updating are mostly about complex facts that cannot be formulated this way. Say, the previous LM holds "The US president is Obama in 2014", and the new knowledge is "The US president is Biden in 2021".
- There are missing important details. Particularly, how do you obtain the entity representations at first? And are there serious justification why entity representation should be the starting point of your formulation? It's a bit vague and unjustified. My suggestion on this is to take the representation of facts instead of the entities as the starting point in your MI formulation.
- The key findings, although useful, are quite trivial and the analysis on the K-Adapter and ERNIE may not be particularly interesting to the audience. My suggestion on this is to either theoretically formulate the KI methods in a unified framework or study more KI methods in this paper.



**Summary Of The Paper:**

This paper presents a method to analyze how knowledge integration methods perform. The proposed approach, graph convolution simulator (GCS) is to understand the KI process by simulating the change of mutual information with graph Fourier transformation as well as interpretable graph attention.

The authors focus on using LM's entity representation as the lens to understand the change of knowledge in LMs during the KI process, and formulate the catastrophic remembering and forgetting via the MI about entity representations. The key idea of the proposed GCS method is to use GFT (graph Fourier transformation) to map entity rep. to the knowledge graph space. And then the authors proposed to use graph convolution to simplify the computation of GFT.

The authors show the effectiveness of GCS by studying two popular KI methods: K-Adapter & ERNIE, then show their pros and cons under different scenarios and finally point out a few findings and future directions.

**Summary Of The Review:**

An interesting approach to study the knowledge integration process, but the proposed method and empirical analysis are both quite limited to particular settings and models.

---

> ### Author Response · Authors · 2021-11-17
> **Reply to Reviewer WDUZ**
>
> Thanks for the review. Currently we are trying to add more experiments to prove that our GCS could provide correct conclusions. We will revise and update the main paper ASAP. Let us try to address your concerns one by one.
>
> 1. Theoretically, we provide some conclusions as well as the proofs in the appendix. We will reorganize the paper to emphasize the theoretical contribution. Empirically, we add experiments to support that our GCS model could provide correct analysis results (see 1.a, 1.b, 1.c). The reason why we don’t include more KI methods is that our method is model-agnostic, which means that it can be deployed on many knowledge-enhanced LMs in principle. And we feel that adding experiments on the method verification directly is more effective compared to analyzing more KI methods. You may find more details about the new experiment results as follows.
>
>     **a. Simulation experiment: verifying GCS and baselines by simulations and comparing results of ERNE/K-Adapter with simulation results**. Specifically, we plan to add a simulation experiment where we set the input of GCS as noisy KG embedding (with different levels of noise) and the output as the groundtruth KG embedding (KGE), and study how probe models interpret the “KI process” (how much KG information is integrated from the noisy KGE to the ground-truth KGE). We select two kinds of baselines to interpret the “KI process”.
>
>     **Representation Analysis** (RA) methods: we calculate the Cosine/Euclidean similarities between two connected entities’ embeddings to see how it changes (Ideally, learning new KG info may make the similarity between entities larger).
>
>     **Probing** method: we design a linear classifier to do link prediction, and observe how its performance changes (Ideally, large positive gaps means that more KG information is integrated.)
>
>     The simulation results that will be updated in paper later show that **our method and RA methods can correctly show the “KI process”**: larger noise ratio would be interpreted with more integrated KG information. And the linear classifier fails to interpret when the noise ratio is large (>10%). Besides, we also set the input as BERT/RoBERTa entity embeddings and output as ERNIE/K-Adapter entity embeddings. We find that **only our GCS can provide meaningful results** (equivalent to noise ratio=5%): a small amount of KG information is integrated. For the baselines, results are not meaningful (RA: similarity becomes larger for K-Adapter and smaller for ERNIE, linear classifier: performance on ERNE/K-Adapter embedding is worse than BERT/RoBERTa).
>
>     **b.** We will report the downstream task performance (entity typing) of BERT and RoBERTa as baselines to complement our previous results, where their performance on downstream tasks are 1-3% worse than ERNIE/K-Adapter (1-2% worse than ERNIE/K-Adapter-drop).
>
>     **c.** We will study the results of the downstream task (entity typing on OpenEntity dataset) to further verify our analysis results. We find that for ERNIE, if we drop the test data that contain entities that have little information integrated during KI (self-loop attention weights>0.1), the performance is slightly better than the original one (1.6%). And if we drop test data that contain entities that integrate much KI information (self-loop attention weights<0.1), the performance is significantly worse than the original one (>20%). The experiment on K-Adapter is still running.
>
> 2. In this paper, we consider a general method without too much side information. The complex KG with different relation types, entity descriptions, multi-hop relations, and time stamps are left for future work.
> 3. We will reorganize the paper to make it more readable, and revise related vague descriptions.
>     a. For the entity representation x_i in terms of LM_1, we input the entity text (label) of that entity into the LM_1 as x_i = LM_1(x_i). Note that since LM gives token-level embeddings, we take the mean value of tokens’ embeddings as x_i.
>     b. The reason why we start from entity representations is based on previous probing works (Hewitt & Manning (2019) and Hou & Sachan (2021)). In their works, they tried to probe linguistic structures (tree or graphs) in LMs. And the representations of nodes are calculated as the mean token embeddings (point 3.a).
>     c. Thanks for the kind suggestion. In our paper, we model the random variables and calculate the MI based in an entity-wise manner (sample number=entity number). We may explore the edge-wise MI in the future.
>
> 4. Indeed, the writing of the paper needs to be improved. We will reorganize the paper and emphasize our theoretical foundation of our method, and focus more on the verification of it empirically. Some findings will be added to the appendix. We plan to update the paper soon as well.
>
> We hope that our response answers your questions and convinces you.

---

> ### Author Response · Authors · 2021-11-23
> **Reply to Reviewer WDUZ**
>
> Dear reviewer,
>
> We would like to thank you for your time, and the opportunity to improve our paper. We have significantly improved the exposition in this paper and we have comprehensively performed all the analyses suggested by the reviewers and we have included the results of those experiments. Overall, we strongly believe that the writing modifications, results of the new experiments and our rebuttal should address many of the concerns raised in this review.  All the changes are summarized in our "Reply to all reviewers".
>
> Thanks,
> Authors of the submission

---

> ### Author Response · Authors · 2021-11-26
> **Reply to Reviewer WDUZ (any new concerns?)**
>
> Dear reviewer WDUZ,
>
>
> Thank you again for your valuable comments! We want to know if your concerns have been properly addressed. If you have any other (unresolved or new) concerns after reviewing our feedback and paper, please let us know. We are very happy to answer them.
>
> Best,
>
> The authors

---

### Official Review · Reviewer_Lpu3 · 2021-11-08

**Correctness:** 2
**Technical Novelty And Significance:** 3
**Empirical Novelty And Significance:** 3
**Recommendation:** 6
**Confidence:** 3

**Main Review:**

Strengths: see "Summary Of The Paper".
Weaknesses:
1. A potential issue about the method: the design of the graph attention network. While I understand most parts of the proof, I find "Graph attention works as edge denoising" to be difficult to understand, especially how "graph attentions are implicitly denosing the edge weights" lead to "we can use attention coefficients in graph attention in graph convolution layer to interpret the KI process". There seems to be a gap in the logic for me. I am confused about this part also because we have multiple (n) graph attention layers, right? So which layer we should use for interpretation? What is the theorical property for using any of the layers? Is it possible that multiple different edge attentions can lead to the same result?
2. The evaluation is weak for me. Although there are interesting findings, but it is difficult for me to judge whether we can trust the results, due to lack of evaluation of the method itself, e.g., comprehensive comparison with baselines either in a qualitative or quantitative way. Although there are theoretical analysis, experiments are still needed since we still have many approximations here, right?
3. It will be really good if there are some ways to aggregating the edge-wise attentions to compute the (decomposed) MI directly (instead of only showing edge-wise results). For example, can we consider the new pipeline with GCNs as a simplied model for which MI could be easily computed (e.g., we only need to compute the MI for the attention layers)?
4. The authors said that "Thus, understanding the KI process requires building probes separately for each of these KI methods." It is not obvious to me why we cannot using model-agnostic explanation methods for black-box models, e.g., [1][2]? This will probe all changes in a unified way, e.g., by perturbing the inputs and see how the perturbation changes the outputs. Maybe the authors could take a look at [2], which also uses MI for general and consistent model interpretation.
5. The authors said that In this work, we consider factual knowledge in the form of triples (vi; r; vj). Thus, it suffices to set G(vi) = Nvi ." I think there are multiple types of edges, so if we set G(vi) to N_vi, we will ignore the relation type, right?
6. The paper presentation may be improved a little to better illustrate the meaning behind the equations. For example, this sentence is difficult for me to understand: "The transformation is a simulation of the KI process, i.e., MI change, and it promise the accuracy." What does Eq. (1) actually mean? Does it verify the correctness for transforming with GFT? How does this align with MI change? Also for the sentence "we analyze the transformation and show that MI change only happens in linear functions of the neural network (Figure 3)." I did not understand it before seeing the proof in Appendix C. Please consider refer to Appendix C here (instead of later before Fig. 3).

[1] Ribeiro, M. T., Singh, S., & Guestrin, C. (2016). “Why Should I Trust You?”: Explaining the Predictions of Any Classifier. https://doi.org/10.1145/1235
[2] Guan, C., Wang, X., Zhang, Q., Chen, R., He, D., & Xie, X. (2019). Towards a Deep and Unified Understanding of Deep Neural Models in {NLP}. Proceedings of the 36th International Conference on Machine Learning, 97, 2454–2463. http://proceedings.mlr.press/v97/guan19a.html

**Summary Of The Paper:**

This paper presents a probe model to analyze how well existing methods integrate knowledge into pretrianed language models. To achieve this goal, the authors first propose a mutual-information-based framework, which is reasonable and provides a unified solution for elegantly understanding whether catastrophic forgetting or catastrophic remembering happen. The authors then propose a method to quickly evaluate the mutual information by using a graph attention network. I really appreciate the theorical foundation that the authors take effort to provide, which is interesting and promising. It is also intriguing how different insights about SOTA models can be collected.

**Summary Of The Review:**

1. Strengths: interesing and important problem, the MI-based framework is reasonable and elegant, most parts of the theoretical proofs are reasonable.
2. Weaknesses: some problems related to method design and/or paper presentation, weak evaluation, insufficient discussion on (or comparison with) some related works

==After rebuttal==

Thank the authors to carefully consider my suggestions. I appreciate the authors' great efforts in improving the paper, e.g., improving method description, adding details missing, and providing three additional numerical experiments. Most of my concerns have been addressed now. However, I can still see that some experimental results are not very good (e.g., Sec. 4.2.3). Considering all these, I would like to raise my score to 6.

---

> ### Author Response · Authors · 2021-11-17
> **Reply to Reviewer Lpu3 (part 1/2)**
>
> Thanks for the constructive comments. The theoretical foundation for the probe model is indeed one of our core contributions. We will soon reorganize the paper to make our contribution clearer. To support our method empirically, we will add experiments with baselines. A new official comment will be posted to indicate the revisions soon. Let us try to address your listed concerns one by one.
>
> 1. This part will be rewritten for clarity. In graph convolutional networks, the number of layers decides the receptive fields when aggregating information from neighbors (see footnote 3). And each layer has different roles for the convolution. For example, given a GCN model (with attention mechanism) that has 2 layers, the attention weight in the first layer shows how a node aggregates information from its neighbors’ features (1-hop aggregation). And the attention weight in the second layer shows how a node aggregates its aggregated information (after the first layer) from its neighbors’ aggregated features (2-hop aggregation). Intuitively, the attention weight in the second layer may indicate how the node aggregates information from its 2-hop neighbors. **Thus, multiple edge attentions can lead to different results**. Regarding KI analysis, knowledge enhanced LMs consider knowledge as triples (without complex multi-hop relations) for integration, and triples only contain entities within 1-hop neighbors. Thus, we can set the number of GCN layers as 1 in practice, and use attention coefficients in that layer for interpretation.
>
> 2. The evaluation of our method is challenging since probing KGs is different from probing small linguistic graphs. The large sparse KG makes it harder to design probe models. Detailed settings and results will be updated in the paper ASAP.  In particular, we plan to add the following experiments:
>
>     **a. Simulation experiment: verifying GCS and baselines by simulations and comparing results of ERNE/K-Adapter with simulation results**. Specifically, we plan to add a simulation experiment where we set the input of GCS as noisy KG embedding (with different levels of noise) and the output as the groundtruth KG embedding (KGE), and study how probe models interpret the “KI process” (how much KG information is integrated from the noisy KGE to the ground-truth KGE). We select two kinds of baselines to interpret the “KI process”.
>
>     **Representation Analysis** (RA) methods: we calculate the Cosine/Euclidean similarities between two connected entities’ embeddings to see how it changes (Ideally, learning new KG info may make the similarity between entities larger).
>
>     **Probing** method: we design a linear classifier to do link prediction, and observe how its performance changes (Ideally, large positive gaps means that more KG information is integrated.) The simulation results that will be updated in paper later show that **our method and RA methods can correctly show the “KI process”**: larger noise ratio would be interpreted with more integrated KG information. And the linear classifier fails to interpret when the noise ratio is large (>10%). Besides, we also set the input as BERT/RoBERTa entity embeddings and output as ERNIE/K-Adapter entity embeddings. We find that **only our GCS can provide meaningful results** (equivalent to noise ratio=5%): a small amount of KG information is integrated. For the baselines, results are not meaningful (RA: similarity becomes larger for K-Adapter and smaller for ERNIE, linear classifier: performance on ERNE/K-Adapter embedding is worse than BERT/RoBERTa).
>
>     **b.** We will report the downstream task performance (entity typing) of BERT and RoBERTa as baselines to complement our previous results, **where their performance on downstream tasks are 1-3% worse than ERNIE/K-Adapter (1-2% worse than ERNIE/K-Adapter-drop)**.
>
>     **c.** We will study the **test results of the downstream task** (entity typing on OpenEntity dataset) to further verify our analysis results. We find that for ERNIE, if we drop the test data that contain entities that have little information integrated during KI (self-loop attention weights>0.1), the performance is slightly better than the original one (1.6%). And if we drop test data that contain entities that integrate much KI information (self-loop attention weights<0.1), the performance is significantly worse than the original one (>20%). The experiment on K-Adapter is still running.

---

> ### Author Response · Authors · 2021-11-17
> **Reply to Reviewer Lpu3 (part 2/2)**
>
> 3. This point is inspiring. If we use the pointwise MI instead, it may be feasible to compute MI directly. We would like to explore it in the future.
>
> 4. We can use the model-agnostic methods. But they are **not suitable for the KG integration scenario** (as also mentioned in point 2.a): the simple probe models cannot work on the link prediction task for the large and sparse KG (i.e., predicting 1 million relations from 1m x 1m relation space.). We appreciate the references about model-agnostic explanation. We would mention model-agnostic methods in related work after we reorganize the paper. Existing knowledge probes try to prompt LMs as the interpretation results, but they can only tell which triples are learned. As for entity-wise integration such as ERNIE, knowledge probes do not perform well about telling you which entities are learned or not. Thus, understanding the KI process (learned entities, learned triples, forgotten/remembered entities) requires building (factual) knowledge probes separately for each of these KI methods.
>
> 5. In this paper, we consider a general method without too much side information. The complex KG with different relation types, entity descriptions, multi-hop relations, and time stamps are left for future work. Under this condition, we can say that knowledge triple information is contained in local KG structure (G(vi)=Nvi).
>
> 6. We would reorganize the main paper and appendix to make it clearer.
>
> We hope that our response answers your questions and convinces you.

---

> ### Author Response · Authors · 2021-11-23
> **Reply to Reviewer Lpu3**
>
> Dear reviewer,
>
> We would like to thank you for your time, and the opportunity to improve our paper. We have significantly improved the exposition in this paper and we have comprehensively performed all the analyses suggested by the reviewers and we have included the results of those experiments. Overall, we strongly believe that the writing modifications, results of the new experiments and our rebuttal should address many of the concerns raised in this review. All the changes are summarized in our "Reply to all reviewers".
>
> Thanks,
> Authors of the submission

---

### Author Response · Authors · 2021-11-22
**Reply to all reviewers**

Dear reviewers,

We thank all the reviewers for the constructive comments, valuable suggestions, and the time spent on our manuscript. Following your comments, we believe we have significantly improved the paper with additional verification experiments and a clear structure. The main improvements we made to the paper are the following.

1. Section 1. We added a paragraph (paragraph 2) to introduce more related works (i.e., model-agnostic explanation methods, prompt).
2. Section 2 and Section 3. First, we reorganize the structure of these two sections and polish them, making it more readable. Second, we emphasize our theoretical contributions by formulating them as Proposition 3 and Proposition 4. Third, we clearly introduce how to use GCS in practice.
3. Section 4. We add two verification experiments (section 4.2.1 & section 4.2.3) as recommended by the reviewers to evaluate our methods with baselines. Some other findings were moved to the appendix.

For each reviewer, we point out the place in the paper that addresses your specific concerns.

For Reviewer Lpu3:
1. Section 3.1 (last paragraph), Appendix D (last paragraph)
2. Section 4.2 (whole subsection)
3. Reply part 2/2 (point 3)
4. Section 1 (paragraph 2 & 3)
5. Section 3.1 (last paragraph)
6. Section 2.2 (paragraph after Theorem 2), Section 2.3 (paragraph after Proposition 3)

For Reviewer WDUZ:
1. Reply (point 1), Section 4.2 (whole subsection)
2. Reply (point 2)
3. Section 2.1 (paragraph 2 & 3), footnote 1
4. Section 2&3, Section 4.2

For Reviewer 4QcM:
1. Section 3.3, Appendix F & G & H
2. Section 1 (paragraph 2&3), Section 2.1 (paragraph 3), Section 3.2 (last paragraph), Appendix H
3. Section 4.2.1
4. Section 2.2 (paragraph after Theorem 2), Proposition 3 & 4, Section 4.2.1
5. Section 4.2.2, Section 4.2.3
6. Reply (point 6)
7. Section 1 (last paragraph)
8. Section 2.1 (paragraph 2)
9. Section 1 (paragraph 2)
10. Section 1 (paragraph 1), Section 2.2 (paragraph after Theorem 2), Appendix K, Appendix J

For Reviewer eLPV:
1. Section 4.2.2
2. Section 4.2.3

For Reviewer pRFd:
1. Section 4.3 (paragraph 3)
2. Section 4.2.1
3. Appendix G & N

We believe our response addresses most of the reviewer concerns. Please let us know if you have any more questions, comments or suggestions.

---

### Decision · Program_Chairs · 2022-01-20

**Decision:**

Reject

**Comment:**

Strengths:
* Theoretical foundation provided to knowledge integration problem
* Findings from the empirical studies are interesting
* Authors dedicated significant time and energy to coordinating with reviewers in the rebuttal period

Weaknesses:
* It is not clear whether the GCS is a suitable approximation for measuring KI. For example, relation types are not supported in the GCS architecture making it unclear whether GCS adequately approximates knowledge integration. As reviewer 4qCM mentions, (X, born_in, Zurich) is very different knowledge from (X, died_in, Zurich). The current formulation only learns co-occurrence between entities rather than relational knowledge.
* Empirical study is limited to two knowledge integration methods (ERNIE & K-Adapter) and only evaluated on entity typing datasets, which are likely to be well-suited for their method which ignores relation information.
* The presentation and takeaways of the results could be clearer. Authors should explain in-depth why experiments that drop knowledge randomly are not suitable baselines.

This paper is promising and the topic explored by the authors is interesting. I think it would benefit from integrating the comments from the reviewers and will make for a strong submission at a future venue.